# Coring of Antarctic Subglacial Sediments

**Da Gong [1,2], Xiaopeng Fan [1,3], Yazhou Li [1], Bing Li [1], Nan Zhang [1], Raphael Gromig [4], Emma C. Smith [4], Wolf Dummann [4], Sophie Berger [4], Olaf Eisen [4,5], Jan Tell [4], Boris K. Biskaborn [4,6], Nikola Koglin [7], Frank Wilhelms [4,8], Benjamin Broy [4], Yunchen Liu [1], Yang Yang [1], Xingchen Li [1], An Liu [1] and Pavel Talalay [1,*]**

1  Polar Research Center, Jilin University, Changchun 130026, China; gongd@jlu.edu.cn (D.G.);
   heaxe@163.com (X.F.); jluyazhouli@163.com (Y.L.); bing@jlu.edu.cn (B.L.); znan@jlu.edu.cn (N.Z.);
   lyc041833@126.com (Y.L.); yangyang2014@jlu.edu.cn (Y.Y.); lixc17@mails.jlu.edu.cn (X.L.);
   liuan18@mails.jlu.edu.cn (A.L.)
2  College of Physics, Jilin University, Changchun, 2699 Qianjin St., Changchun 130012, China
3  State Key Laboratory of Cryospheric Science, Northwest Institute of Eco-Environment and Resources,
   Chinese Academy of Science, Lanzhou 730000, China
4  Alfred-Wegener-Institut für Polar- und Meeresforschung, Postfach 120161, 27515 Bremerhaven, Germany;
   gromigr@uni-koeln.de (R.G.); emma.smith@awi.de (E.C.S.); wdummann@uni-koeln.de (W.D.);
   sophie.berger@awi.de (S.B.); Olaf.Eisen@awi.de (O.E.); jan.tell@awi.de (J.T.);
   Boris.Biskaborn@awi.de (B.K.B.); frank.wilhelms@awi.de (F.W.); Benjamin.Broy@awi.de (B.B.)
5  Department of Geosciences, University of Bremen, 28359 Bremen, Germany
6  Alfred Wegener Institute Helmholtz Centre for Polar and Marine Research, Telegrafenberg A45,
   14473 Potsdam, Germany
7  Federal Institutefor Geosciences and Natural Resources (BGR), Geozentrum Hannover, D-30655 Hannover,
   Germany; nikola.koglin@bgr.de
8  Department of Crystallography, Geoscience Centre, University of Göttingen, 37073 Göttingen, Germany
*  Correspondence: talalay@jlu.edu.cn; Tel.: +0086-431-8850-2546

**Abstract:** Coring sediments in subglacial aquatic environments offers unique opportunities for research on paleo-environments and paleo-climates because it can provide data from periods even earlier than ice cores, as well as the overlying ice histories, interactions between ice and the water system, life forms in extreme habitats, sedimentology, and stratigraphy. However, retrieving sediment cores from a subglacial environment faces more difficulties than sediment coring in oceans and lakes, resulting in low yields from the most current subglacial sediment coring methods. The coring tools should pass through a hot water-drilled access borehole, then the water column, to reach the sediment layers. The access boreholes are size-limited by the hot water drilling tools and techniques. These holes are drilled through ice up to 3000–4000 m thick, with diameters ranging from 10–60 cm, and with a refreezing closure rate of up to 6 mm/h after being drilled. Several purpose-built streamline corers have been developed to pass through access boreholes and collect the sediment core. The main coring objectives are as follows: (i) To obtain undisturbed water–sediment cores, either singly or as multi-cores and (ii) to obtain long cores with minimal stratigraphic deformation. Subglacial sediment coring methods use similar tools to those used in lake and ocean coring. These methods include the following: Gravity coring, push coring, piston coring, hammer or percussion coring, vibrocoring, and composite methods. Several core length records have been attained by different coring methods, including a 290 cm percussion core from the sub-ice-shelf seafloor, a 400 cm piston core from the sub-ice-stream, and a 170 cm gravity core from a subglacial lake. There are also several undisturbed water–sediment cores that have been obtained by gravity corers or hammer corers. Most current coring tools are deployed by winch and cable facilities on the ice surface. There are three main limitations for obtaining long sediment cores which determines coring tool development, as follows: Hot-water borehole radial size restriction, the sedimentary structure, and the coring techniques. In this paper, we provide a general view on current developments in coring tools, including the working

principles, corer characteristics, operational methods, coring site locations, field conditions, coring results, and possible technical improvements. Future prospects in corer design and development are also discussed.

**Keywords:** subglacial aquatic environments; hot-water access borehole; sediment corers; water–sediment interface

## 1. Introduction

Subglacial sediments in aquatic environments are found at the seabed beneath ice shelves, subglacial lake beds, and at the base of ice streams. They contain important paleoenvironmental and paleoclimatic records [1,2] that provide information linking the overlying ice with the aquatic system [3], as well as information on the biodiversity and the evolutionary processes of living organisms under subglacial conditions [4,5] and the subglacial sedimentation processes [6]. Moreover, subglacial sediment cores may even contain more information beyond the age of the ice core [7]. To obtain scientific information and verify hypotheses, in situ exploration and sampling of subglacial sediments [8,9], i.e., subglacial sediment coring, is needed.

### 1.1. Where Subglacial Sediments Occur and Detection Methods

Antarctic subglacial aquatic sediments occur mainly in three environments, as follows: Sub-ice shelves, sub-ice streams, and subglacial lakes [10–12]. Figure 1 shows the scheme of subglacial sediment occurrences in Antarctica. However, the sediment depositional processes are still unknown unless sediment cores are collected and sedimentary sequences are analyzed [1,13].

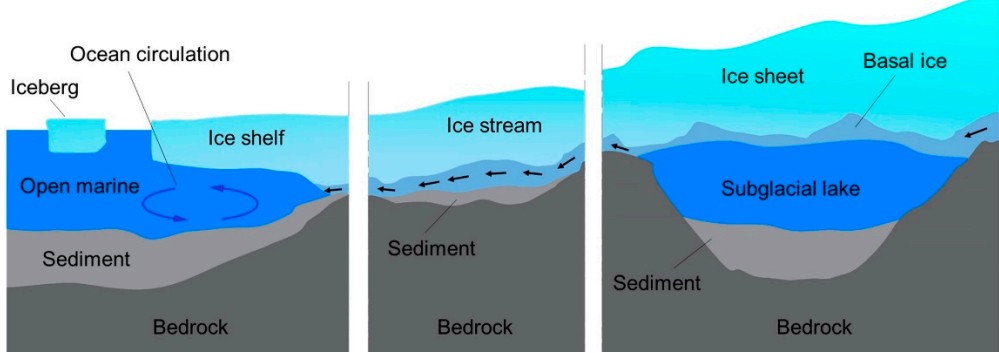

**Figure 1.** Schematic diagram of exist circumstances of the Antarctic subglacial aquatic sediments.

The sediments beneath Antarctic sub-ice aquatic environments are located and characterized mainly by seismic sounding survey methods that can also provide information on the water cavity thickness [6,8,9]. Along with determining sub-ice bathymetry, seismic reflection data can show the occurrence of soft sediments and even indicate sediment depth and its degree of coarseness or fineness [6]. All of the above information is critical for sediment coring equipment selection, development, and applications [6].

### 1.2. Reasons for the Interest in Coring and Requirements

Compared with lake and ocean sediments, subglacial aquatic sediment deposits are from different and more diverse source materials, from the transport and melting of the overlying ice, the hydrological flow [12], and biological and volcanic activities, for example.

Sediments beneath ice shelves are most likely deposited from two main sources, as follows: The open marine environment and basal debris from the ice shelf base near the grounding zone [10].

Therefore, research on the sub-ice-shelf sediment cores is mainly focused on interpreting presence or absence of the ice shelf in the geological record, inferring the interaction between the sub-shelf and open-water marine environments [3], and obtaining boundary information about sub-ice shelf sedimentation [14,15]. Sub-ice shelf sediment research also aims to study the microbial ecosystem and carbon cycling in these unique environments [15].

Considerable interest in sub-ice-stream sediments focuses mainly on rheology and its importance to ice streaming and the mass balance and stability of the modern Antarctic ice sheet [16]. One of the main aims in obtaining such information is to determine when the ice sheet last decayed, which is critical to assessing the present-day risk of ice sheet collapse and consequent sea-level rise [8,17]. Such information cannot be provided by ice cores, as these are restricted to the age of the ice itself (~1 million years) [7], whereas the sedimentary records show that the last time central West Antarctica was ice-free ~1 million years ago [18]. Sediment cores can provide paleontological and geochemical data for paleo-climate research and infer the time the west Antarctic ice shelves collapsed [19,20].

Over 400 subglacial lakes exist underneath the Antarctic Ice Sheet [21], where the temperature of ice reaches the pressure melting point. Subglacial lake sediment research efforts are more focused on biological questions [9,22,23] and efforts are focused on the following: (i) What form of microbial life exists in Antarctic subglacial lakes [1,9,22,23]; (ii) the post-Pliocene history of the Antarctic ice sheet [1,9]; (iii) paleoenvironmental and paleoclimatic records that can determine the ice sheet dynamics [1,9]; and (iv) the biodiversity and the evolutionary processes of living organisms under unknown habitat conditions [4,9].

Based on these sediment research interests and the requirement for subsequent core analysis, there are two basic requirements for subglacial sediment sampling, as follows: (i) The water–sediment interface should be undisturbed for biological analysis [9,23] and (ii) there should be minimal disturbance and deformation of the layer deposition structure in sediment samples for research based on time scales [9,10], for example, sequence stratigraphic analysis [13].

*1.3. Sediment Sampling Restrictions and Difficulties in Subglacial Aquatic Environments*

1.3.1. Specific Working Conditions

Before sediment coring, an access borehole should be drilled using the hot water drilling method. The diameter of the borehole may range from 10 to ~60 cm [24] and the depth from a few to thousands of meters [4,22]. When the hot-water drill penetrates the ice base, the borehole water level will adjust to the local hydrological level, achieving pressure equilibrium [3]. The sediment coring tools are then deployed, first through the air filled and then water filled parts of the borehole, through the ocean or lake water column if present, to reach the sediment, which is sampled and retrieved to the surface as sediment core samples [8].

The access borehole closes at rates of between 3 to ~6 mm/h [25,26], i.e., it refreezes at an L113 change to a rate that depends on the ice temperature surrounding the borehole, the water pressure, the initial borehole size, and the water composition [24]. Sub-ice instruments can only be deployed and retrieved during the window of opportunity between the time the borehole is drilled/reamed and the time it refreezes to a certain size [24]. Otherwise, instruments can become stuck in the borehole. Therefore, sediment sampling devices are always cylindrical shaped and primarily designed and selected based on the hot water borehole diameter and water pressure [27]. The stabilizing frame [28] and overhanging trigger system [29] used in lake and ocean sediment coring are not applicable due to their size (Figure 2).

In certain subglacial lake settings, any contamination from the surface is strictly forbidden, thus sediment coring tools should be sterilized and cleaned prior to being deployed into the borehole [30,31].

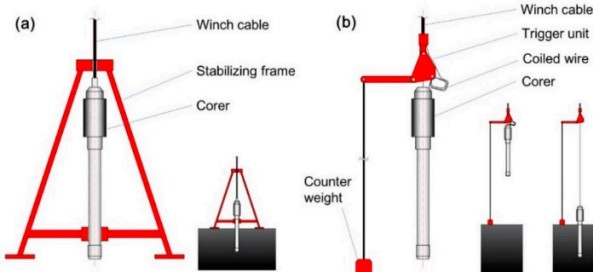

**Figure 2.** Structure and function schemes of the conventional corers with (**a**) a stabilizing frame and (**b**) an overhanging trigger system; the radial sizes (150–600 cm) of these corers are far larger in diameter than can enable access to hot-water drilled boreholes (10–60 cm).

### 1.3.2. Surface Operation and Control Facilities

The ANDRILL (Antarctic Geological Drilling) project on Ross Ice Shelf [32] used a basic winch and tower, a high-strength cable (with or without power transmission capacity), a control unit, and a platform for sub-ice sediment coring. The cable, winch, and tower were designed to manage loads from between 1500 and 4500 kg [33]. The operational height between the tower top pulley to the borehole surface is generally between 3–6 m, while heights of more than 10 m can be achieved using a hydraulic powered crane [34]. Assembly and disassembly operations of the sampling devices are usually executed by manual rather than by automatic auxiliary equipment.

### 1.3.3. Complicated Sediment Structure

Subglacial sediments are more heterogeneous compared to normal aquatic sediments, due to the multiple sources of their deposits throughout their histories [12]. The basic composition of such sediments is generally the same as marine sediments (silt, clay, sand fractions, gravels, for example.) but the structure and mechanical properties are strongly influenced by the overlying ice layer dynamics [10]. In some drilling sites, the shear strength of cores obtained can be more than 70 kPa at a depth of 2 m [35], and the surface biological debris (2–4 mm) and deeper interbedded gravel (4–64 mm) become the main factors preventing corers from going deeper [6]. Therefore, coring tools equipped with additional kinetic energy units have been developed to penetrate deeper [11,23].

Although many attempts have been made to collect subglacial sediment samples, only a few studies have successfully recovered sediment samples from beneath Antarctic subglacial settings. Coring results have generally shown low success rates and penetration performance. In this paper, we review the design and characteristic structure of representative sediment coring tools developed for Antarctic subglacial aquatic environments. We provide some constructive suggestions and our expectations regarding the technologies are also included in order to express new ideas on how to improve current coring techniques and corer designs.

## 2. Current Subglacial Aquatic Sediment Corers: Method, Working Principle, Design, and Applications

Only a handful of studies have successfully recovered sediment samples from Antarctic subglacial aquatic environments [8]. The coring tools used have been commercial products (e.g., UWITEC corer) or purposely designed based on the requirements of hot water drilling boreholes and the parameters of subsidiary facilities (e.g., gravity corer developed by the Alfred Wegener Institute). For specially designed corers, the coring methods and parameters have been chosen based mainly on the following points: (1) The sampling length aims, (2) sediment properties and thickness that can be roughly derived from seismic reflection surveys [6], and (3) compatibility between coring methods and sediment types [8,33] (e.g., coring requirements for ocean and lake sediments). All corers may be classified into two categories based on their research purpose. The water–sediment interface sampling corer (0.3–2 m core barrel) is used for microbial or organic matter research and the long-tube corer (1.5–6 m core

barrel) is used to recover the longest possible time-scale sediment sequence [8,35,36]. Moreover, corers may also be classified based on their working principle, including single gravity percussion corers, hammer percussion corers, vibration liquefaction corers, and piston corers. Here, we review the coring tools that have been custom-designed for, or that have already been deployed in Antarctic subglacial sediment sampling. They are classified based on their working principle and presented according to structural complexity. Drilling sites (Figure 3), locations, and borehole conditions are also presented.

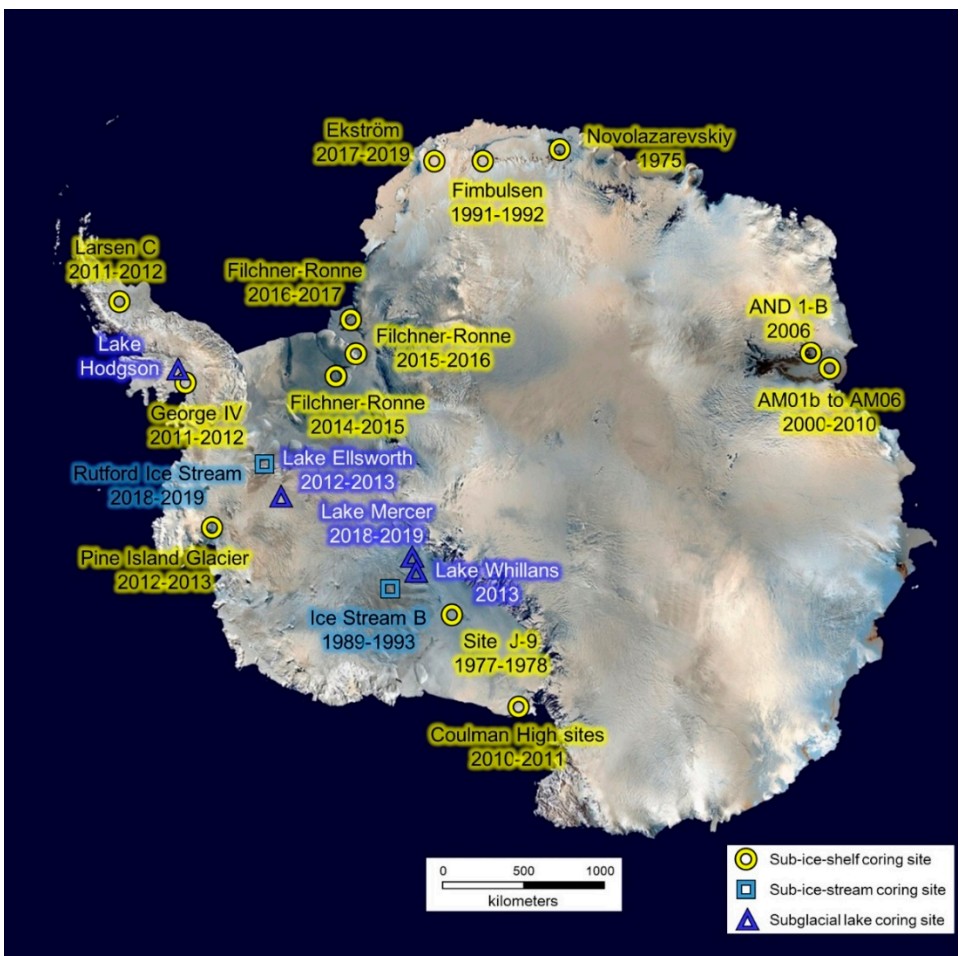

**Figure 3.** Site locations of subglacial sediment coring in Antarctic.

## 2.1. Gravity and Gravity Piston Corers

Gravity coring is the most widely used method to obtain sediment cores from subglacial aquatic environments [8]. Almost all hot water drilling projects are equipped with a gravity corer if the coring plan involves subglacial sediment coring. The reasons for this are as follows: (1) This corer is associated with higher success rates on most types of sediments (from soft sediments to hard clay) compared to other corers; (2) its simple structure simplifies designs and lowers costs, and (3) its adjustable dead weight makes this corer adaptive to different winch systems and cables [8,37].

The working principle of the subglacial gravity corer is generally the same as the conventional gravity corer except for the balance beam trigger system [29], which consists of a dead weight, the core tube (double or single tube), an optional cutter, an optional core catcher, a check valve mechanism, an optional tube piston mechanism, and a cable connector (Figure 4). Due to the borehole size restriction, the radial the trigger system must be removed (Figure 2b). As a result, the balance beam rod trigger system cannot be used, thus a surface winch releases the corer at a distance 5–22 m above the sediment surface [35]. Thus, the corer 'free-falls' and pushes the core barrel to penetrate through the sediments

by its gravitational potential energy [37]. The penetration depth of a gravity corer depends on the net weight of the coring assembly, the diameter of the core catcher and core liner, the hydrodynamic properties of the coring assembly (e.g., water drag resistance), and the physical properties of the sediment [38].

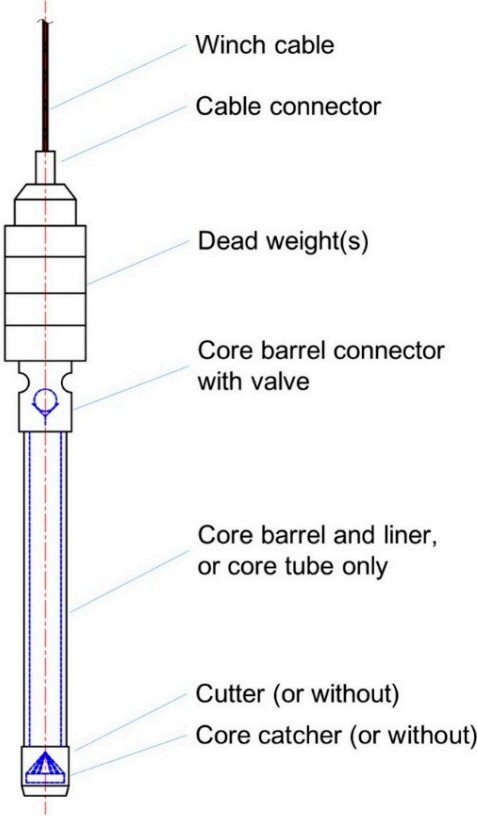

**Figure 4.** General layout of subglacial sediment gravity corer.

Published data shows that there are mainly two gravity corer types depending on sampling aims, including (1) long tube sequence gravity corers, which will obtain sediment cores 1–6 m long, and (2) light corers, which will obtain undisturbed surface water–sediment interface core samples.

### 2.1.1. 'Benthos Model 2171' Gravity Corer

With the possible exception of the sediment corer used on the Novolazarevskiy ice shelf [39] (which was not identified), the first subglacial sediment gravity corer is the 'Benthos model 2171' used at Site J-9 (82° 22′ S, 168° 38′ W) as a part of the Ross Ice Shelf Project [40]. During the austral summer of 1977, 58 short cores with lengths of up to 122 cm were collected at Site J-9 [40,41]. Samples were 400 km from the open water of the Ross Sea, where the ice thickness was about 420 m, the marine water column was 237 m, and the distance between the water surface to the seafloor was 597 m [40]. The access borehole was drilled through the ice shelf using a flame-jet drill developed by the Browning Engineering Corporation and produced a hole 30–80 cm in diameter [42]. This corer was also used to obtain 30–60 cm cores from beneath the Fimbulsen Ice Shelf in 1991–1992 [8,43]. The 'Benthos model 2171' gravity corer can be equipped with a 2.4 m long liner (ID 67 mm), has a total weight of 110 kg, and can be equipped with a 3 m core barrel [8,40].

### 2.1.2. 'Wintle' Corer

The 'Wintle' gravity corer was purposely built for sub-ice-shelf sediment sampling. It is 150 cm in length and its maximum diameter is 12 cm [6,44]. This corer was first used at site AM02 (69°42.8′ S,

72°38.4′ E) on the Amery Ice Shelf, where it collected a 144 cm core ~80 km south of the floating ice shelf front in the 2000–2001 drilling season, as part of the Australian National Antarctic Research Expeditions (ANARE) Amery Ice Shelf Oceanographic Research (AMISOR) program [6]. This 144-cm core was X-ray radiographed to examine for any sedimentary structures and ice-rafted debris (IRD) granules (2–4 mm) and pebbles (4–64 mm) were counted for every 5 cm interval downcore [44]. In the 2003–2004 season, a second 47 cm core was collected from site AM01b (69°25.86′ S, 72°26.77′ E), which is 50 km west of the AM02 site. Four other cores between 60 and 124 cm long were obtained from sites AM03 to AM06 in the 2005–2006 and 2009–2010 seasons [10].

### 2.1.3. AWI Gravity Corer

On 16 October 2006, the ANDRILL team used a gravity corer at the AND-1B site (77°53′22″ S, 167°5′22″ E), after drilling the hot water borehole (~40 cm in diameter) and prior to the deployment of the sea riser [35]. This project aimed to recover the water–sediment interface and a few decimeters of sediment below the surface. Eight coring attempts were made and 7 cores with lengths of up to 53 cm were recovered [35]. The small (~80 kg, maximum diameter 22 cm) gravity corer (Figure 5a) was designed and built by AWI (Alfred Wegener Institute). The corer was fitted with a plastic core barrel that was a UWITEC standard plastic liner that was 1.0 or 1.5 m long and 59.5/63 mm in ID/OD and was without a cutter and core catcher [37]. The weight of the corer can be adjusted by adding or reducing lead weight loops. A self-contained tension triggering valve mechanism (lock unit) was installed at the top of the corer. The valve is kept open during the lowering and penetrating steps; it then closes once enough instantaneous tension is exerted on the lifting bar (Figure 5b,c). After passing the corer through the access borehole, it is lowered to 5–25 m above the sea-floor and stabilized for about 1–2 min. Then the surface winch is switched to a 'free-wheel' mode and the corer penetrates into the sediment by its gravitational potential energy. After penetration, the surface operator exerts a sudden force to the cable to close the sealing lid. The core with the intact water–sediment interface is kept under vacuum in a space formed by the sealing lid, the core tube, and sediment [35].

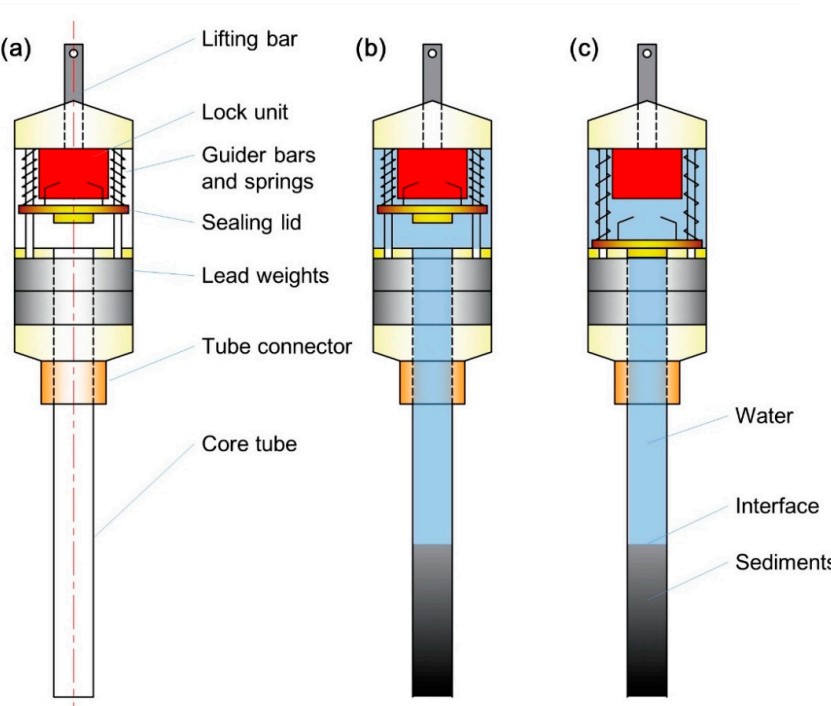

**Figure 5.** (**a**) Principle structure of the AWI gravity corer, (**b**) the corer working states during penetrating, and (**c**) the corer working state of lifting with core.

The AWI gravity corer was again deployed by the ANDRILL team at Coulman High beneath the Ross Ice Shelf between November 2010 and January 2011, as part of preparations for a planned drilling project targeting an Early Cenozoic sequence [37]. In total, 28 cores with lengths of up to 129 cm were obtained from four hot water drilling sites [37].

The AWI gravity corer was used in several hot water boreholes (~40 cm) at the Ekström Ice Shelf between November 2018 and January 2019, as part of the Sub-EIS-Obs project [45]. Seven gravity coring attempts were made but only one succeeded in obtaining a water–sediment interface sample with a 46 cm core (Figure 6b) from the drill site, where the ice thickness was 332 m and the seafloor was located ~700 m from the ice surface.

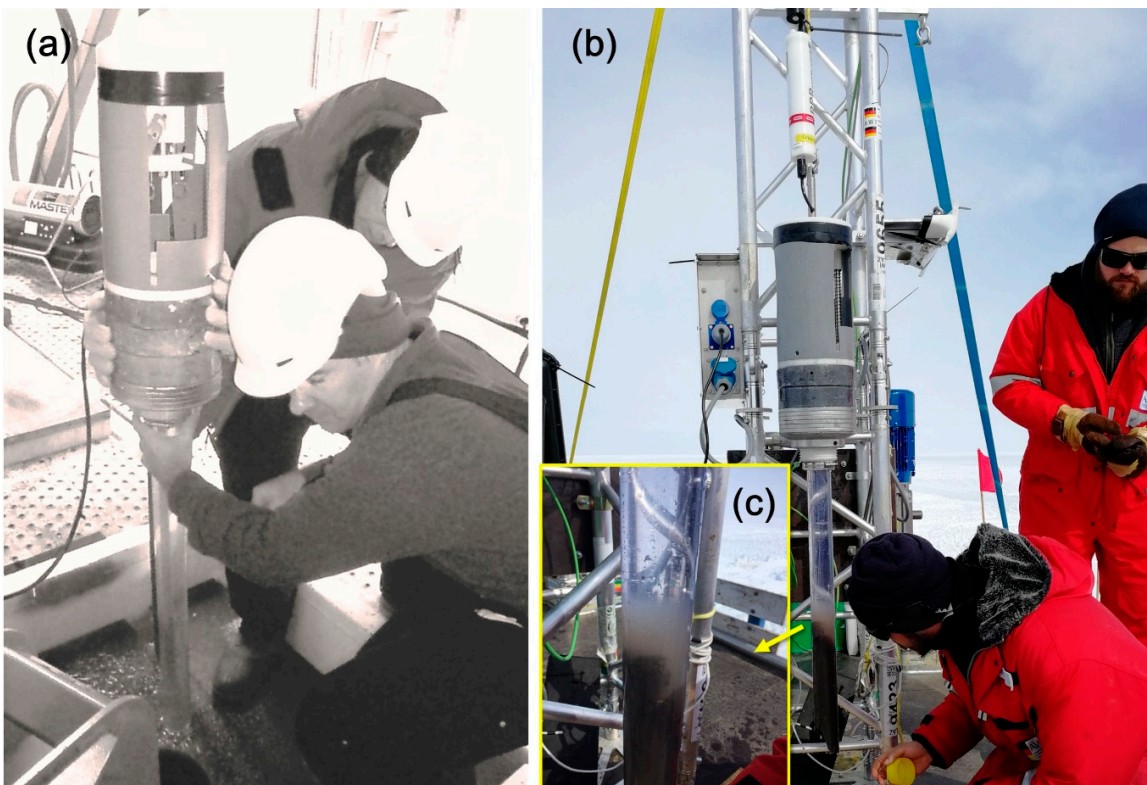

**Figure 6.** The AWI gravity corer applied at (**a**) site AND-1b [35] (Reproduced with permission from Terra Antartica Publication, 2007), at (**b**) the Ekström ice shelf with (**c**) undisturbed water–sediment interface ((**b**,**c**) photographs were taken by Y. Li).

### 2.1.4. UWITEC Gravity Corer and NIU/UWITEC Multi-Corer

The UWITEC (an Austrian engineering company) gravity corer is designed to sample soft and watery surface sediments with the aim of preserving the water–sediment interface [46]. This gravity corer has a streamlined shape and is equipped with an automatic core catcher (Figure 7). It weighs 5–8 kg and has a maximum OD of no more than 150 mm (without the ball core catcher). The core tube dimensions are standard, with a UWITEC liner with ID/OD measurements of 59.5/63 mm and a length of 80–120 cm. The corer can be released to 'free-fall' several meters above the sediment surface. The line-ball core catcher and valve flap automatically seal the tube's lower and upper ends, respectively, when coring is completed. Additional dead weights can be added if needed, with each tube circular ring weighing 4 kg [46]. The UWITEC gravity corer has successfully retrieved a surface sediment core from a coring site (72°00.259′ S, 68°29.022′ W) at Lake Hodgson [47], where the ice thickness is 3.7 m and the distance between the ice surface and the sediment is 93.4 m [22].

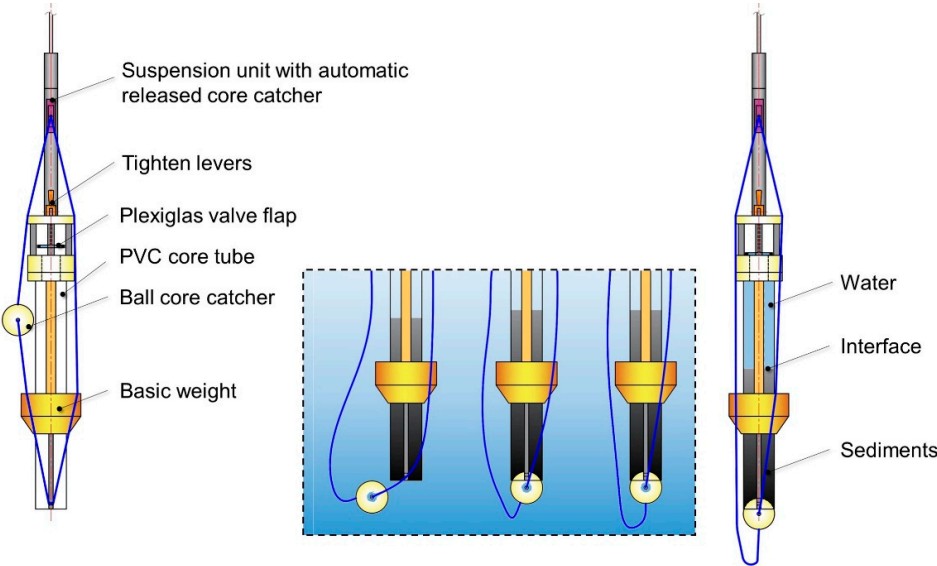

**Figure 7.** Structure and sealing sketch of the conventional UWITEC gravity corer [46].

After being modified by Northern Illinois University (NIU), the corer was renamed the NIU/UWITEC gravity multi-corer (Figure 8) [1]. This multi-corer is a combination of three standard UWITEC gravity corers, installed on a trisection frame (Figure 8a). Its working principle is the same as the standard UWITEC gravity corer, but it can take three replicate cores simultaneously after being self-triggered upon striking the bottom sediment. This corer has been used in at least two sites since it was designed, including at the Subglacial Lake Whillans (~60 cm diameter access borehole) site by the WISSARD (Whillans Ice Stream Subglacial Access Research Drilling) team in January 2015 (Figure 8b), where it collected eight good cores with lengths between 20 and 40 cm [48,49]. However, seven attempts were unsuccessful (e.g., corer was stuck in the borehole) because the corer assembly is relatively large, is not streamlined, and is quite light [1,50]. Recently, the corer was deployed at Subglacial Lake Mercer Whillans (~60 cm diameter access borehole) by the SALSA (Subglacial Antarctic Lakes Scientific Access) team in the 2018–2019 season, and at least six water–sediment interface cores were obtained [51].

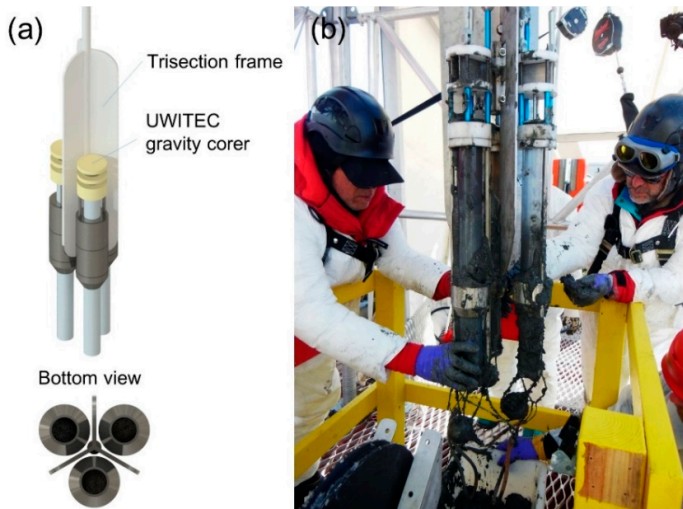

**Figure 8.** NIU/UWITEC multi-gravity-corer: (**a**) 3D structure scheme and multi-core obtained from (**b**) Subglacial Lake Whillans [50] (photograph was taken by M. Beitch, courtesy of the National Science Foundation).

### 2.1.5. BAS/UWITEC Gravity Corer

This short corer (Figure 9) was developed cooperatively by BAS (British Antarctic Survey) and UWITEC, with the aim of collecting surface sediment cores in sub-ice-shelf settings. It consists of a valve head, which also functions as a dead weight, a single liner, and an orange-peel shaped core catcher. This corer was used during 2014–2015 and 2015–2016 field seasons by BAS to sample sub-ice-shelf sediments. Approximately 30 cm diameter access boreholes were drilled by a hot water drilling system through the Filchner-Ronne Ice Shelf, where ice thickness ranges between 750–891 m. In total, seven cores were obtained with a combined total length of 265 cm in the 2014–2015 season and at least three cores of up to 55 cm were obtained in the 2015–2016 field season [52]. Three cores with lengths up to 75.5 cm were obtained from BAS drill sites on the Filchner-Ronne Ice Shelf during the 2016–2017 field season, where ice thickness ranges between 597–615 m and water column thickness ranges between 440–643 m.

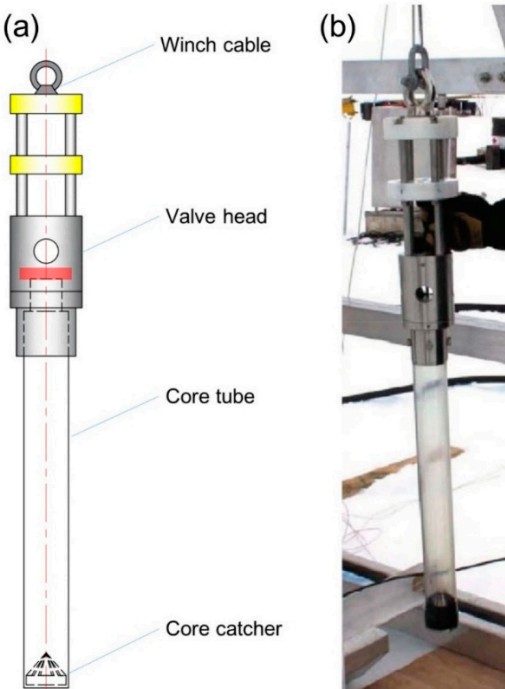

**Figure 9.** The BAS/UWITEC gravity corer: (**a**) General structure and (**b**) used in the field [8] (Reproduced with permission from Royal Society, 2016).

### 2.1.6. SLE Gravity Corer

To obtain long sediment core samples through a tether without an electric power supply, the SLE (Subglacial Lake Ellsworth) team proposed a heavy gravity corer (Figure 10) with a 270 kg head weight, enabled for preparative sterilization and cleaning [31]. This corer is equipped with a single tube core barrel that is 3.7 m long and has a 60 mm inner diameter to achieve a reasonable balance between corer diameter and corer weight. The combination of a long and heavy tube with a heavy load was designed for deeper penetration. The entire corer is driven into the sediment by only gravitational potential energy [8].

This corer was planned to be used in the subglacial Lake Ellsworth in the 2012–2013 season by passing through an access borehole approximately 36 cm wide and ~3000 m deep [36] to obtain old sediments. The seismic reflection data analysis showed that the lake's sediment is highly porous with a low density, very similar to material found in the deep ocean floor. The sedimentary sequence thickness was estimated to be at least 2 m [53]. Unfortunately, the hot-water drilling failed to access the lake [54] and the SLE gravity corer has not been applied.

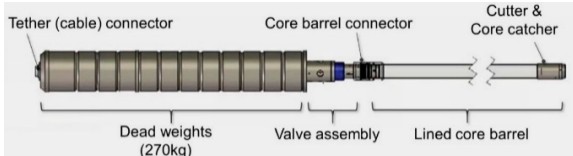

**Figure 10.** Schematic of SLE gravity corer [8] (Reproduced with permission from Royal Society, 2016).

### 2.1.7. WHOI Gravity Corer

The WHOI (Woods Hole Research Institute) gravity corer (Figure 11) is the largest tool in SALSA's lineup, with a total length of 9.1 m. Its purpose is to take up to ~6.0 m core samples by its heavy weight (~1130 kg). At subglacial Lake Mercer in 2018–2019 season, the WHOI corer was deployed through a ~60 cm diameter access borehole and it made contact with the sediment bed at a depth of 1084 m. Two cores with lengths of 70 cm and 170 cm were obtained with this corer, setting a new length record in subglacial lake sediment coring [55,56].

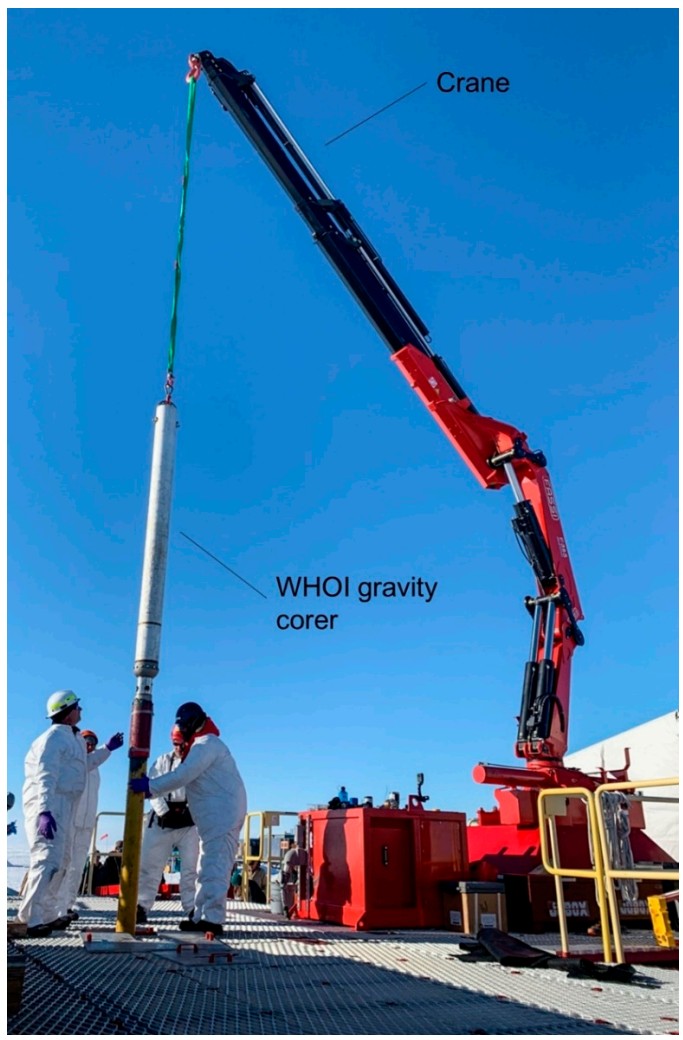

**Figure 11.** The WHOI gravity corer [34] (photograph was taken by K. Kasic, SALSA Education & Outreach).

### 2.2. Hammer/Percussion Corers

Subglacial aquatic sediment hammer corers or percussion corers (Figure 12) were developed to achieve deeper penetration than the gravity coring tools. The main difference between the hammer/percussion corer and the gravity corer is the power supply used to achieve core tube

penetration. Gravity corers penetrate into sediments through single-use, gravity-based energy (dependent on dead weight and release distance); in contrast, hammer/percussion corers are equipped with a hammer/percussion head that provides continuous impact energies to push the cutter and barrel into sediment. The hammer/percussion corer design consists of a double-tube core barrel structure, including a steel core barrel to sustain impact force and the liner to retain the sediment core samples. The hammer/percussion heads are driven by an ice surface cable manually operated, using electric power transferred through an armed cable or batteries integrated in the corer body [8].

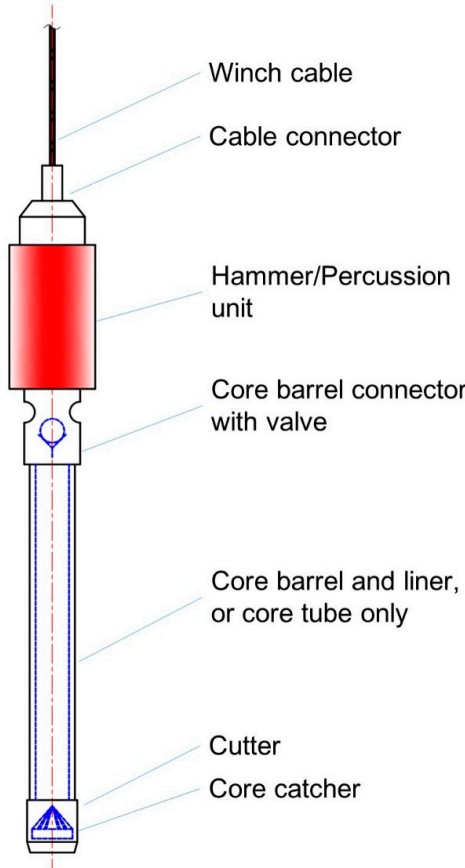

**Figure 12.** General layout of subglacial sediment hammer/percussion corer.

2.2.1. BAS/UWITEC and UWITEC Manually Operated Percussion Corer

The BAS/UWITEC percussion corer was developed by the BAS in collaboration with UWITEC [8]. It consists of a hammer unit, the valve head, and a core barrel unit (Figure 13). The maximum diameter of this corer is 14 cm and the 3 m long version with three hammer weights makes a total of ~130 kg. The corer was designed to pass through a ~30 cm wide access borehole. The double-tube core barrel unit consists of an outer barrel (ID/OD 64/70 mm) made of stainless steel, a 3 m long UWITEC PVC liner (ID/OD 59.5/63 mm), a double layered orange peel core catcher, and a cutter nose (ID/OD 59/74 mm). This corer can be deployed to core 6 m long sediment samples if it is equipped with a 6 m double-section core barrel. Lowering, lifting, and downhole coring procedures can be manually accomplished by a single winch cable through surface operations.

This BAS/UWITEC manually operated percussion corer was used during the 2011–2012 austral summer in the hot-water drilling project at Larsen C and George VI ice shelves. A total of 1160 cm of sediment cores were recovered with a maximum penetration of 290 cm [14,57,58]. BAS recovered several cores using the BAS/UWITEC corer, including (i) cores with lengths up to 115 cm from beneath Pine Island Glacier during 2012–2013 [59,60]; (ii) a 135.5 cm long core from FNE2 drilling site during the 2016–2017 season; (iii) a 152 cm long core from FNE3 drilling site during the 2016–2017 season; (iv)

a ~10 cm long coarse gravel core from beneath Rutford Ice Stream drill site, where ice thickness was 2152 m during the 2018–2019 season [61], respectively. This corer was deployed at several hot-water boreholes (~40 cm in diameter) at the Ekström ice shelf between November 2018 and January 2019 as part of the Sub-EIS-Obs project [45]. Eight coring attempts resulted in 7 cores with a total length of ~826 cm and single core lengths of up to 200 cm.

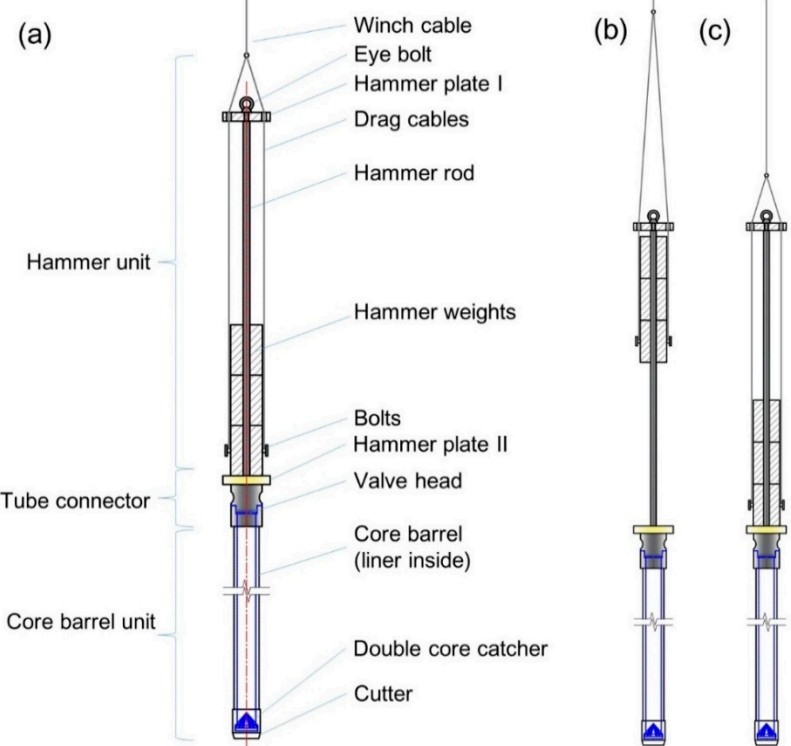

**Figure 13.** (**a**) Structure of the BAS/UWITEC manually operated percussion corer: (**b**) Hammer lifting state and (**c**) hammer lowering state.

Another UWITEC percussion corer deployed in the Antarctic is the UWITEC KOL Kolbenlot percussion piston corer, a commercial product used for lake sediment coring, working at depths of no more than 140 m [46]. This 2 m long corer is manually operated from the ice surface [62]. Three overlapping 2 m long cores with a total penetration depth of 3.76 m were obtained at the Subglacial Hodgson Lake core site (72° 00.256′ S, 68° 29.022′ W), where the ice was 3.7 m thick and the lake bed was located 93.4 m below the ice surface [22].

### 2.2.2. JLU Manually Operated Hammer Corer

The JLU (Jilin University, China) manually operated hammer corer was used at the Neumayer Station III in Antarctica to obtain a long core sample (>100 cm) with an undisturbed water–sediment interface. The corer consists of a hammer unit, the main frame, and a core barrel unit (Figure 14). The maximum diameter of this corer is 23 cm and it weighs ~130 kg, including its 45 kg hammer. The single core tube (UWITEC liner) measures 59.5/63 mm (ID/OD) excluding the outer barrel, and the cutter (ID/OD 59/66 mm) with the core catcher and the water–sediment interface protector are attached at the lower end of the core tube. The coring procedure is generally the same as that for the BAS/UWITEC, except for an additional step included to preserve the water–sediment interface. The water–sediment interface can be kept at an undisturbed state in the liner, even after the corer is lifted to ice surface. The JLU manually operated hammer corer was deployed at the Ekström ice shelf between November 2018 and January 2019 as part of Sub-EIS-Obs project [45] and it successfully obtained a 128 cm long core with a well-preserved water–sediment interface (Figure 15).

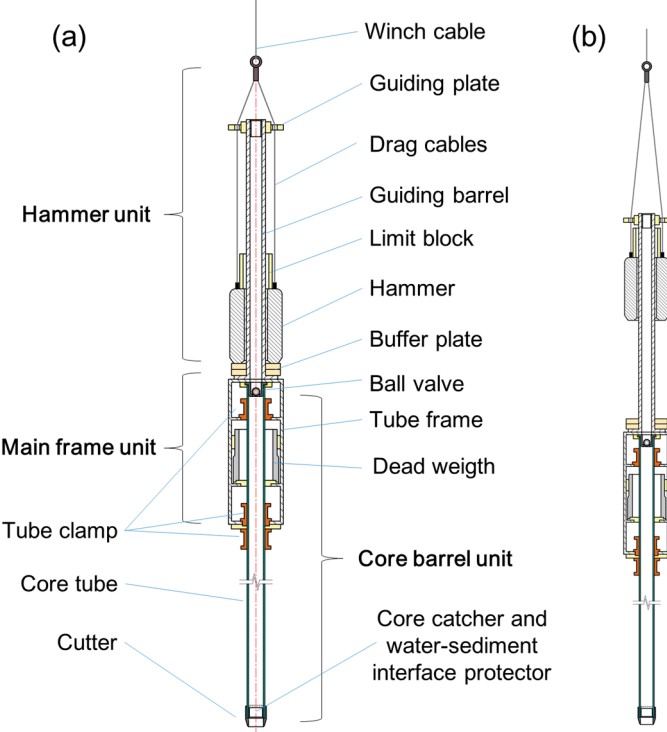

**Figure 14.** (**a**) Structure of the JLU manually operated hammer corer and (**b**) the hammer lifted state.

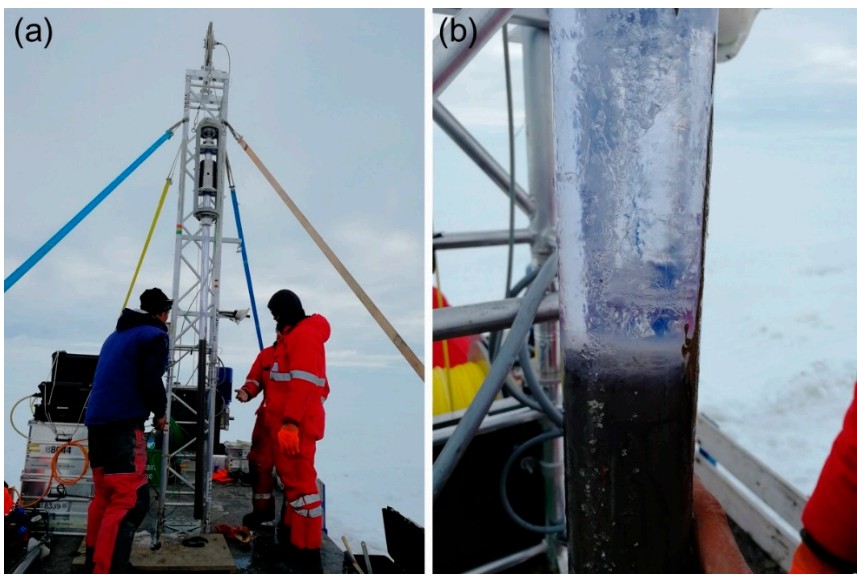

**Figure 15.** (**a**) The JLU manually operated hammer corer lifted with 130 cm core sample and (**b**) the well-preserved water–sediment interface (photographs were taken by Y. Li).

### 2.2.3. NIU Percussion Corer

The SLW (Subglacial Lake Whillans) hydraulic percussion corer (Figure 16) or NIU (Northern Illinois University) percussion corer was designed by S. Vogel and DOER-Marine to obtain deeper, hard, and over-consolidated sediment cores, such as subglacial till cores [1,33]. The corer is designed to be sterilized and it measures a total length of ~14 m and has a ~220 mm maximum OD, including a 5 m long core barrel with ~100/141 mm ID/OD. It is powered by a hydraulic system (motor, piston, for example) that includes a ~900 kg percussion drop mass; a power, control, and telemetry unit; and a

lifting assembly. A smart Kevlar-reinforced fiber-optic cable (4500 kg load ability) connected to the multipurpose winch is used to deploy the corer.

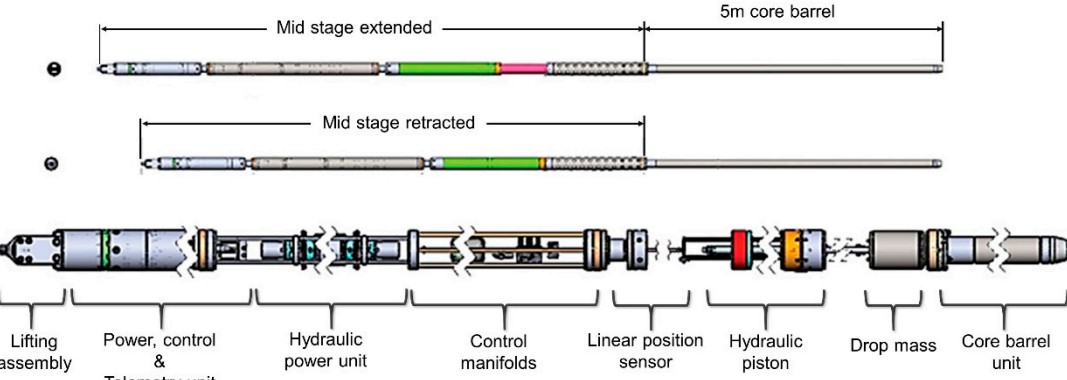

**Figure 16.** Structure drawing of the NIU percussion corer [33] (Reproduced with permission from WISSARD, 2019).

To ensure that the pullout strains between core barrel and sediments do not exceed the cable's maximum tension capacity, the corer can enable a 'hydraulic helping' mode. First, the hydraulics can be commanded to force pressurized water down to the gap between liner and core barrel, then, the water exiting via jet holes in the core cutter head is forced up the outside of the core barrel to decrease the friction between the corer and sediments [1,8,33].

This corer was deployed to the hot water drilling site at the Subglacial Lake Whillans (84.240° S, 153.694° W) in late January 2013 within the WISSARD project (Figure 17). Unfortunately, the corer's working modes failed to work as designed due to a malfunction in the surface smart winch. Eventually, used as a gravity corer, it was finally able to collect a 0.4 m long gravity sediment core [1].

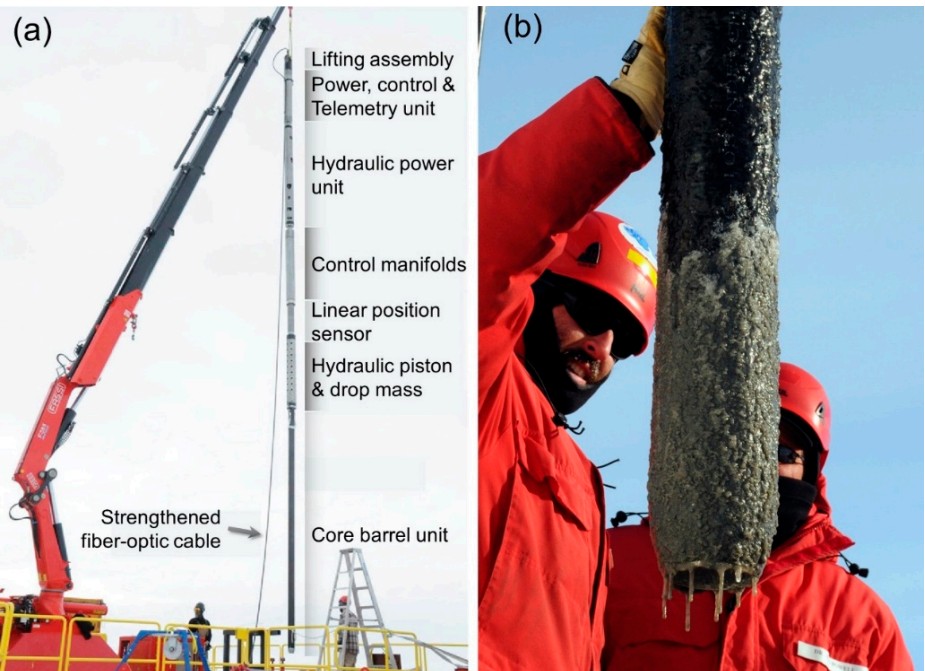

**Figure 17.** (**a**) The NIU percussion corer deployed at Subglacial Lake Whillans [8] and (**b**) the core barrel after cored [63] (photograph was taken by Reed Scherer, courtesy of the National Science Foundation).

### 2.2.4. SLE Percussion Piston Corer

The SLE percussion piston corer (Figure 18a) was designed and manufactured by UWITEC and BAS [7]. The corer is designed to be sterilized and has a length of 5.8 m and a maximum diameter of 20 cm [7,8]. This corer was the first 'real-time visual corer' to study the Antarctic subglacial sediment environment. It consists of five main parts, including a control interface, the hammer, the core barrel, the piston, and the piston-clutch [8]. The control interface contains power converters, control electronics, and communication modems. The hammer has a linear motor drive hammer mechanism sealed in the pressure chamber. The core barrel (3-m long version), liner, cutter (hardened steel) and double-layered orange peel core catcher were borrowed from the UWITEC corer design [7]. The corer's clutch can prevent the piston from moving further to deform the sediment core when the core barrel is full or when sediment penetration has ceased [8]. Cameras are integrated with the piston, pressure chamber, and near the clutch; thus, operators working on ice surface can monitor and control the coring procedure through a conducting tether (Figure 18b,c). The coiled umbilical cable can compensate for the distance changes occurring between the control interface unit and the hammer unit during coring [8].

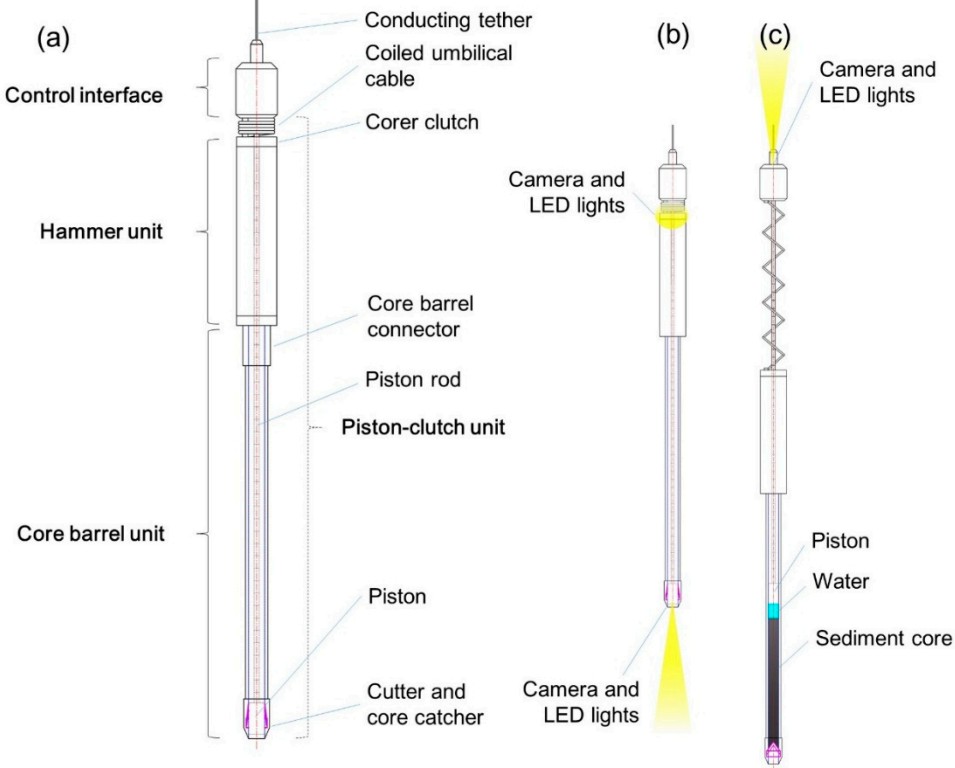

**Figure 18.** (**a**) The SLE percussion piston corer general structure layout, (**b**) lowering state, and (**c**) lifting state with core obtained.

This corer has been specifically and progressively designed to implement technical innovations in subglacial aquatic sediment sampling, such as controlled coring through real-time monitoring, the ability to obtain a long core with a well-preserved water–sediment interface, and the separation between the control cabin and the power cabin. Unfortunately, this corer has not yet been deployed due to the same reason [54] cited for the SLE gravity corer.

### 2.3. Piston Corers

Piston coring is one of the more common seafloor or lake-floor sediment sampling methods. From the perspective of power supply modes, no corer deployed in the Antarctic subglacial sediment setting has ever been purely a 'piston corer'. Instead, the piston units have always used additional

technological units such as push coring, gravity coring, percussion coring, and vibrocoring. All of these reduce the friction between the inside wall and the sediments, and assist in the evacuation of displaced water from the top of the corer [11]. The working principle of the piston-equipped subglacial sediment corers is generally the same as that of the open water sediment corer (Figure 19).

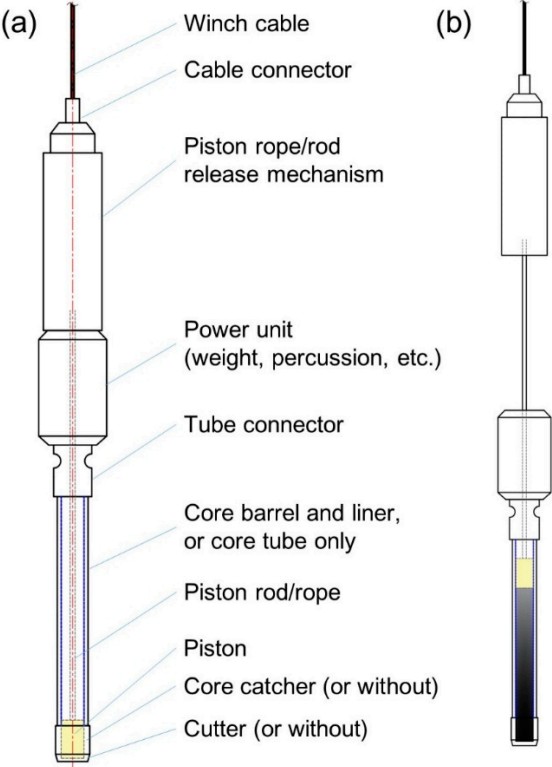

**Figure 19.** (**a**) General layout of subglacial sediment piston corer and (**b**) corer state with core obtained.

In this review, we have listed only the Caltech piston corer and WISSARD piston corer as corers that can be classified as 'pure piston' corers. Other corers that include a piston unit have been classified based on their characteristic penetration power resource methods (e.g., the SLE percussion piston corer).

### 2.3.1. Caltech Piston Corer

The Caltech piston corer was designed and built by a California Institute of Technology (Caltech)-led team that conducted sediment coring beneath Ice Stream B (at camp UpB), West Antarctica [16]. This corer design was derived from the marine piston corer, which is 6 m long with a 50 mm liner ID. It is equipped with a cutter, a metal core catcher, and a plastic liner fitted into the steel core barrel. Several cores of up to 400 cm long [19] were recovered through about 50 hot-water-drilled access holes with an average depth of ~1030 m at Upstream B, where sediments are deposited about 600 m below sea level [8]. More detailed information about this corer's design and its working principle were not published.

### 2.3.2. WISSARD/Caltech Piston Corer

The WISSARD piston corer (Figure 20) was based on the design of the previous Caltech piston corer. It was designed and built by the University of California Santa Cruz to collect non-deformed sediment cores at the bottom of subglacial Lake Whillans [8,33]. This borehole corer is equipped with a 3 m long core barrel with a 58 mm ID [1]. The corer is designed to be sterilized and is mainly made of stainless steel. On its upper stage, there is a tube containing a coiled Kevlar release cord that is used to dead drop the corer [8]. Several external guiding plates at the upper end of the corer increase the

possibility of vertical insertion. This corer needs to be released precisely at a few meters above the lake floor, triggered by a wireline messenger [8]. The piston unit stops at the sediment surface during penetrating and lifting. This corer can obtain cores without significant compression or disturbance. The WISSARD piston corer was used to recover an 80 cm long core during the second phase of scientific operations at the subglacial Lake Whillans drilling site in late January 2013 [1].

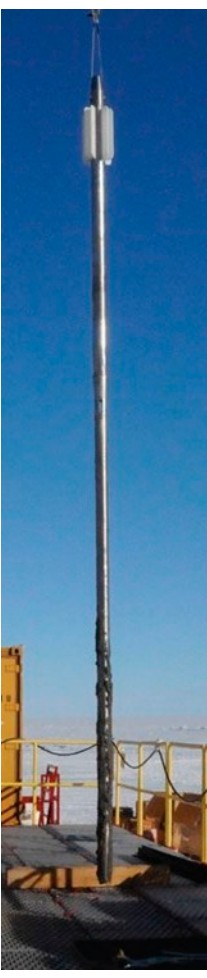

**Figure 20.** WISSARD/Caltech piston corer with sediments packed at lower part [8] (Reproduced with permission from Royal Society, 2016).

## 2.4. Push Corers and Mini-Corers

### 2.4.1. SLE Mini-Push Piston Corer

The water–sediment interface is the prime area for detecting microbial life, especially in the ultra-oligotrophic lakes such as Lake Ellsworth [23]. To be able to collect such samples, a mini-push piston corer was designed and built by the SLE team to obtain the crucial water–sediment interface in this subglacial lake. This corer is mounted on the front (lower end) of the Lake Ellsworth Probe, which is sterilized before deployment, and samples are capped upon retrieval (Figure 21) [7]. The corer is 39 cm long and is designed to obtain a composite core sample up to 22 cm long and contains the water–sediment interface [23]. All metallic parts of the corer are constructed from titanium (grade V), which, along with the nonmetallic components, ensure the validity of trace Fe analysis of all water samples [23]. The SLE probe includes a video camera and lighting system to enable continuous monitoring of the area below the probe. After the sediment surface is sighted and assessed visually by the surface operator, the corer is then deployed to start coring [8]. The coring deployment by itself

takes approximately 3.5 min with a penetration speed of 1 mm/s [23]. To simplify the description, the whole coring process is divided into two phases, the penetrating (Figure 22b) phase and the ball valve sealing phase (Figure 22c). During the penetrating phase, the lead screw drives the whole corer assembly to penetrate the sediment. After the corer reaches the target penetration depth, the corer assembly automatically initiates the sealing phase, i.e., the lead screw continues to rotate, moving the ball drive plate down, which rotates the ball valve by 90° and seals the core tube end [23].

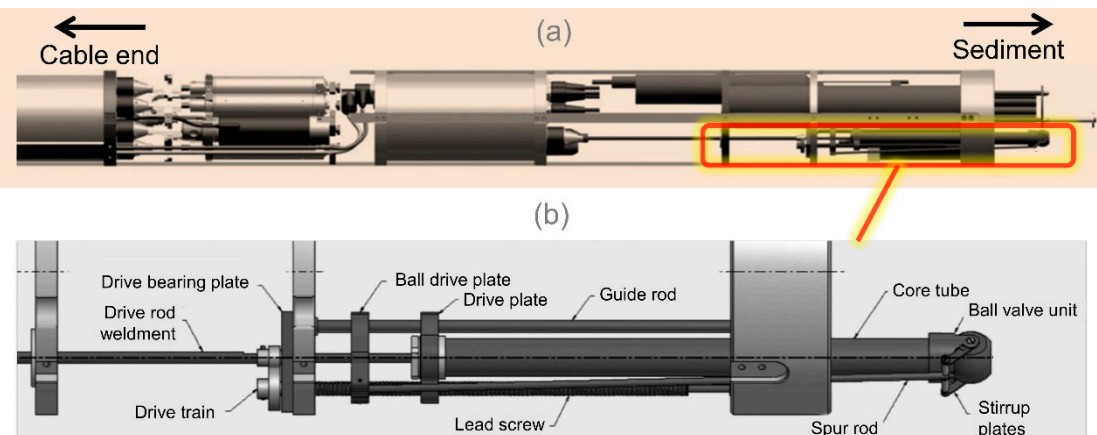

**Figure 21.** (**a**) Position of the SLE mini-push piston corer integrated in the Lake Ellsworth Probe and (**b**) the mechanical structure of the SLE mini-push piston corer [23] (Reproduced with permission from Royal Society, 2018).

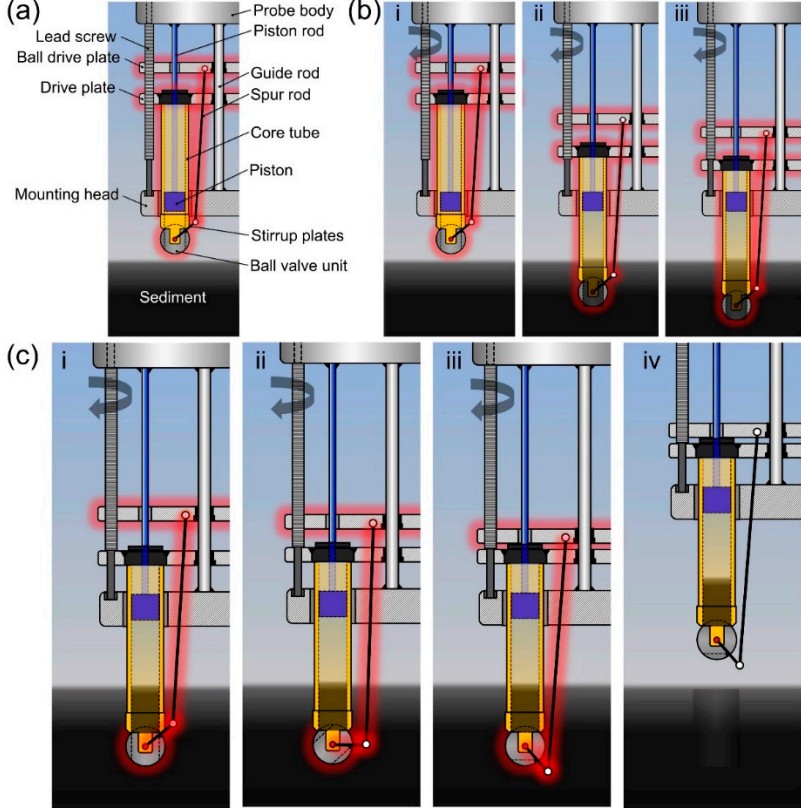

**Figure 22.** SLE mini-push piston corer: (**a**) Mechanical layout of the corer parts, (**b**) penetration procedure, and (**c**) ball valve sealing procedure.

The piston keeps a static position relative to the probe body (mounting head) during the whole coring process, thus preventing fluid intrusion from the outside [23]. Since the core water freezes during the probe's ascent back through a cold (approximately −18 °C) air-filled section, the piston was designed with a volume compensation mechanism (Figure 23) that can compensate for the water volume change when water inside the core tube freezes.

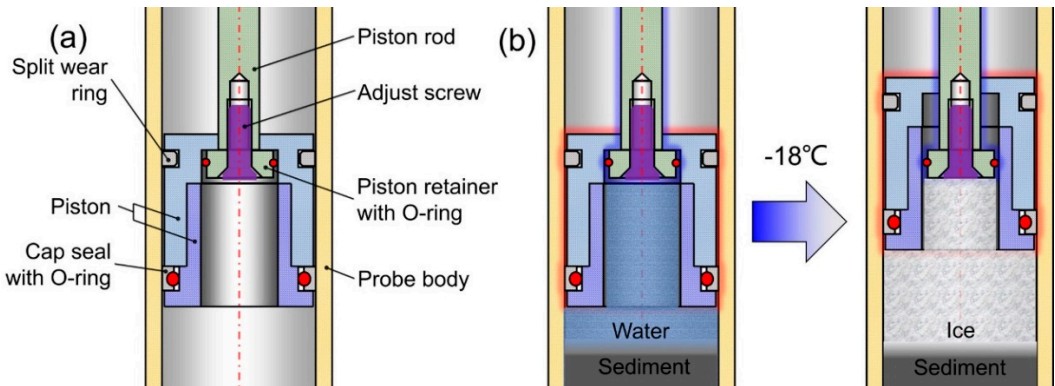

**Figure 23.** (**a**) Mechanical structure layout of the piston unit. (**b**) Principle and procedure for how the piston compensates volume change of frozen water.

This corer passed testing on artificial laboratory sediments as well as deep-marine sediments (1020 m water depth). The results show that this corer can obtain samples in all the sediment types assessed, and the piston unit and other mechanisms worked well [23]. Unfortunately, the BAS hot-water drill failed to access Lake Ellsworth in 2013 and the corer has not been tested in the Antarctic.

### 2.4.2. ANDRILL String Mounted Push Corer

A push corer inner tube assembly was deployed at the AND-1B site (77°53′22″ S, 167°5′22″ E) by the ANDRILL drilling team in October 2006, using a sea riser and a drill string which were lowered to within a few meters of the seafloor. Different from the AWI gravity corer previously deployed at same site, the push corer is not cable-operated but is installed on the drill string and is directly 'pushed' into the sediment without rotation or the use of drilling mud. This corer is 1.6 m [35] long and it is attached to the end of an 83 mm diameter drill bit [15]. Using this method, four cores, varying in length from 33 to 156 cm, were collected by the drill string-mounted ODP (Ocean Drilling Program) liner. Most cores were lost during the lifting operation, resulting in only 216 cm of sediment cores obtained from sediments at depths of between 0.70 m and 10.18 m [35]. The reason we list this non-cable-suspended corer here is to provide a results contrast with the AWI gravity corer that was deployed at the same access borehole.

### 2.5. JLU Self-Synchronous Vibrocorer

As a consequence of the penetration depth limit of sediment coring tools used for Antarctic subglacial aquatic sediment sampling, a narrow bore corer with a 'self-contained power unit' is required to obtain deeper cores [11,22].

Vibrocoring is a simple and efficient technique for obtaining high–quality sediment core samples in a variety of configurations and sizes [64,65]. A vibrocorer is sometimes considered as a high-frequency and low-amplitude hammer corer; however, this view is incorrect because the working principles are different. The vibrocorer utilizes the 'vibrating liquefaction effect' [65] of water-saturated sediments, i.e., the sediments adjacent to the core tube can be re-orientated to a 'shear strength lost' status, then the core tube can be pushed into the sediment using the weight of the vibrocorer itself [66,67].

The first Antarctic subglacial sediment vibrocorer, designed and built by JLU, aimed to address the technical gaps involved in obtaining long sediment cores. To cope with the access borehole size

limitation problem (the main reason that vibrocoring had not been applied yet), the JLU vibrocorer is equipped with a cylindrical vibro-head that gives the whole corer a maximum outer diameter of 27 cm. There are two versions of the JLU vibrocorer. One is designed as a cable power supply corer (Figure 24a) for the hot water drilling project of CHINARE (Chinese National Antarctic Research Exploration) and the other is designed as a self-powered corer for the Sub-EIS-Obs hot water drilling project of AWI (Figure 24b). Both of these corers have been tested in the laboratory with satisfactory results. Sediment cores with lengths of up to 3 m contained sandy mud and hard clay [27].

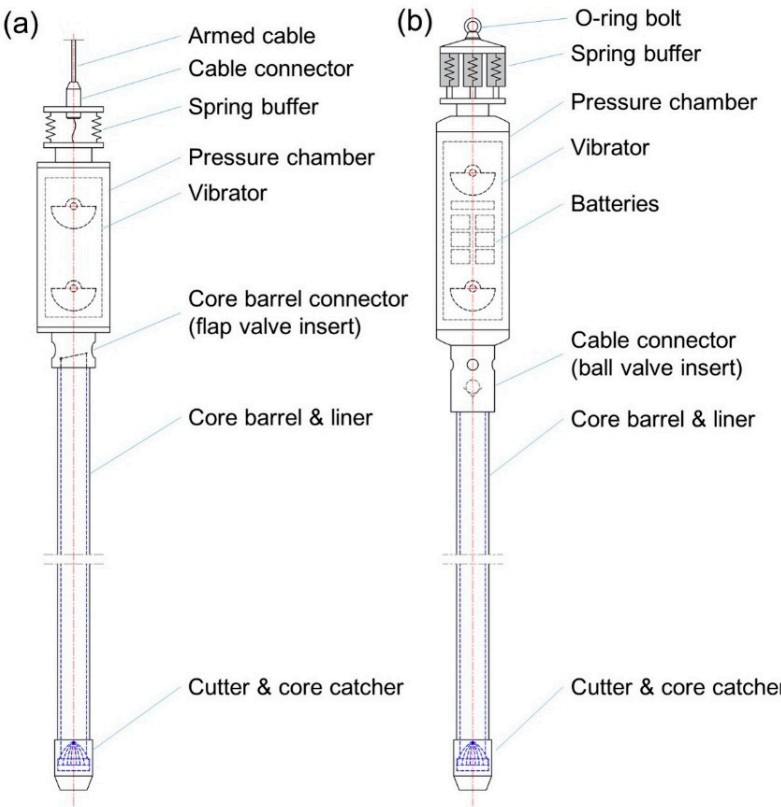

**Figure 24.** The JLU vibrocorer: (**a**) the cable powered CHINARE version and (**b**) the AWI version with self-contained batteries.

The JLU vibrocorer was first applied underneath the Ekström Ice Shelf f during the 2017–2018 season as part of Sub-EIS-Obs exploration project. The general structure of the JLU vibrocorer is shown in Figure 24b. The core tube unit was originally designed to collect 3 m or 6 m cores with a diameter of 106 mm. The cutter OD is 130 mm and the whole corer is 4.7 m long and weighs 256 kg, including the 3 m core barrel. A self-synchronous vibrator first was designed and tested. This can produce a vibration frequency of 50 Hz and a maximum acceleration of ~300 m/s$^2$. The vibrator is powered by self-contained batteries and the vibration can be started on a time-delay. During the 2017–2018 season, the corer was deployed through the ice shelf twice but no core was retrieved because the load cell on the tripod frame top was broken. Without accurate load parameters, the corer was deployed relying only on information on cable depth and the time to set the vibration delay switch. The corer was able to reach the sediment layer; however, because it fell on its side at the seafloor, the only sediments collected were those that adhered on the core catcher and the spring buffer unit.

To improve the corer's penetration ability, three aspects have been modified as follows: (1) It was equipped with a more powerful vibro-head, powered by standard Makita tool batteries, that can work on intermittent vibration modes; (2) the custom-made core barrel (ID/OD is 106/124 mm with liner inside) was replaced with the conventional UWITEC core barrel (ID/OD is 59.5/70 mm with liner inside), thus reducing the cutting area by half; and (3) a tube guider was added. The improved

JLU vibrocorer was redeployed underneath the Ekström Ice Shelf f during the 2018–2019 working season, also as part of Sub-EIS-Obs exploration project. Unfortunately, after being deployed on its first vibrocoring run, it was not possible to lift the corer from the sea floor. At that time, the system had been deployed for around 20 min, to await the start of the vibration. After more than 12 h of continuous attempts to lift the corer, using increasing loads on the winch, the 3 mm diameter Dyneema cable broke with a tension load of 1500 kg (i.e., the specified maximum load) and the deployed corer was lost. By that time, the cable had frozen to the borehole wall, which freezes at a rate of around 5 mm/h (i.e., 1 cm/h in diameter).

### 2.6. DBTS Drill (Corer)

After the Russian team reached the surface of the subglacial Lake Vostok in 2012 and 2015, there has been anticipation in collecting the bottom sediments because these may contain unique information regarding the geology, natural environment, and climate changes in Central Antarctica [30,68]. In 2017, a cable-suspended DBTS (Dynamically Balanced Tool String) coring tool was proposed by the Saint-Petersburg Mining University. It has been especially designed to be sterilized for sediment coring in the subglacial Lake Vostok through a deep 5G borehole [30].

This corer is lowered to the lake surface by a hauling assembly using a carrying cable connected to a surface winch (Figure 25a). The hauling assembly works as a transporter and contains a small winch insert with heating elements. After the hauling assembly reaches the working area 1–2 m above the lake surface, the corer is lowered down by a 1 mm diameter thin cord coiled around the capstan of a small winch. Meanwhile, the heating elements operate to mitigate against the liquid column re-freezing. After the corer touches the sediment layer, the lower drill pipe with the drill bit starts rotation and drilling [30].

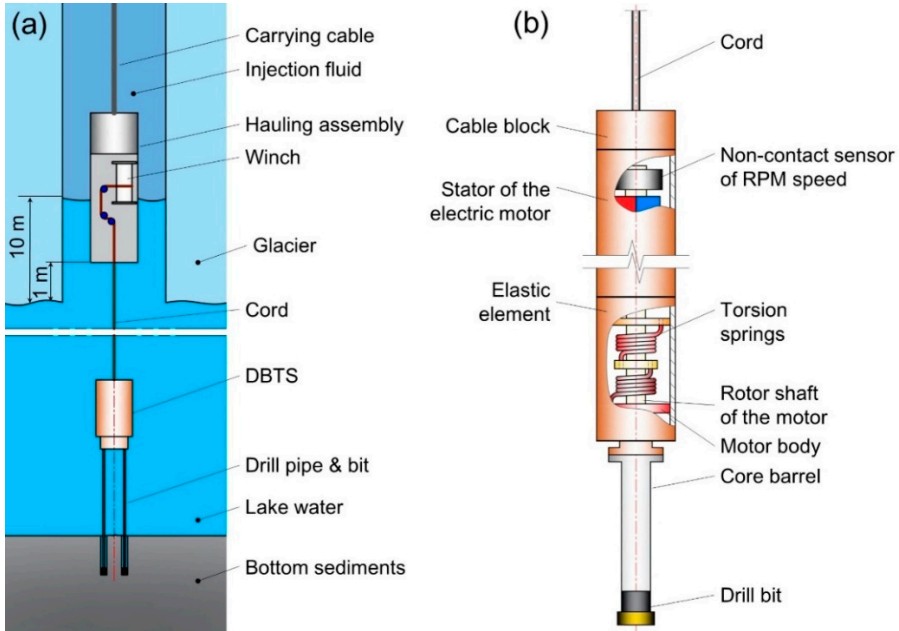

**Figure 25.** General layout of Lake Vostok sediment sampler: (**a**) Operation scheme of the sampling deployment and (**b**) the sampler driven by the DBTS.

The main feature of this corer is the DBTS system (Figure 25b), a two-mass oscillatory electromechanical system with reciprocating and rotating motions with two degrees of freedom (rotation and penetration). A drilling system with a DBTS can perform rotary drilling without any external support on the borehole walls to bear the counter torque [30].

This corer has two optional operational modes, a self-driven mode and a controlled mode, which is operated from the surface. In the self-driven mode, all the operations are carried out automatically and there is no communication between the research module and the surface personnel [30]. Laboratory tests of the DBTS motor drive have been carried out on simulated and physical models, and have confirmed the feasibility of the DBTS system. It has been suggested that a DBTS with a diamond drill bit can even drill through a solid brick [30].

*2.7. Unidentified Corer*

During the 20th Soviet Antarctic Expedition in 1975, one 28 cm core was collected under the Novolazarevskiy Ice Shelf at a site (70°14.2′ S, 11°51.4′ E) with 375 m thick ice. The ice surface was 203 m above sea level [39]. However, the depth of the sea bottom and the core type were not published.

## 3. General Review of Antarctic Subglacial Aquatic Sediment Coring

Data related to subglacial aquatic sediment coring in the Antarctic are summarized in Table 1 and Figure 3. The list starts from the first short 28 cm long core obtained beneath the Novolazarevskiy Ice Shelf to the longest ~400 cm longs cores from the Up-B site. With the exception of the corer used to collect the sediment beneath the Novolazarevskiy Ice Shelf, the methods used for coring in Antarctic subglacial aquatic sediments can be classified as the following: Gravity coring, piston coring, push coring, hammer/percussion coring, and vibrocoring. The different coring methods have been developed based on borehole access and conditions, surface equipment, and water depth. Details on sediment conditions are very hard to determine before coring because current seismic sounding survey methods can only derive general sediment information, such as depth from the ice surface, estimated layer thickness, and comparative acoustic properties (between ocean/lake sediments) [23,69]. Obstructive materials such as biological debris, granules (2–4 mm), and gravels (4–64 mm) can be found until a downhole camera is deployed or a core is obtained.

*3.1. Coring Objectives and Their Respective Coring Methods*

Most mainstream research is interested in the two following main coring objectives in subglacial aquatic sediments: (i) To obtain cores with an undisturbed water–sediment interface that is considered the prime site for detecting microbial life [31] and (ii) to obtain longer cores with a minimum of disturbance and deformation for research based on sequence stratigraphy [9,10].

3.1.1. Coring Methods for Obtaining Undisturbed Water–Sediment Interface

For this coring objective, the surface sediment must enter the core tube unhindered, therefore, the cutter or tube end must be designed with a cylindrical opening (Figure 26) and the corer must preserve the core during lifting. Three cable-suspended coring methods with four specific features have been deployed, as follows: A gravity corer with an open tube end and a cap for sealing in the sediment (Figure 27b) (e.g., AWI gravity corer) or an external line-driven ball core catcher with a valve flap (Figure 27c) (e.g., UWITEC gravity corer); a push corer with a rotary ball cutter and piston unit (Figure 28c) (e.g., SLE mini-push corer); and a hammer corer with a cutter, core catcher, interface protector, and ball valve part (Figure 29d) (e.g., JLU manually operated hammer corer). During lifting, all the above methods and features enable a water vacuum above the interface. Note that an open-end structure sometimes results in a lost core during lifting, especially when sediment is soft and the vacuum above the sediment is unable to hold the core in place. Furthermore, a core tube end without a cutter can sometimes be broken if the corer is 'free-fall' released towards seafloor, where it hits gravel (Figure 30).

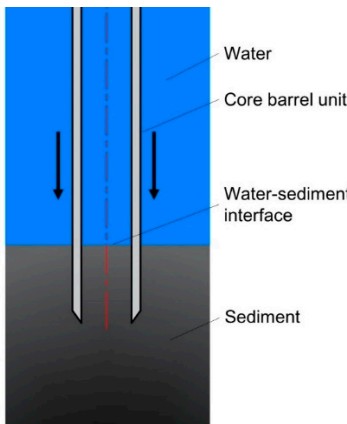

**Figure 26.** Penetration layout of corer with open tube end.

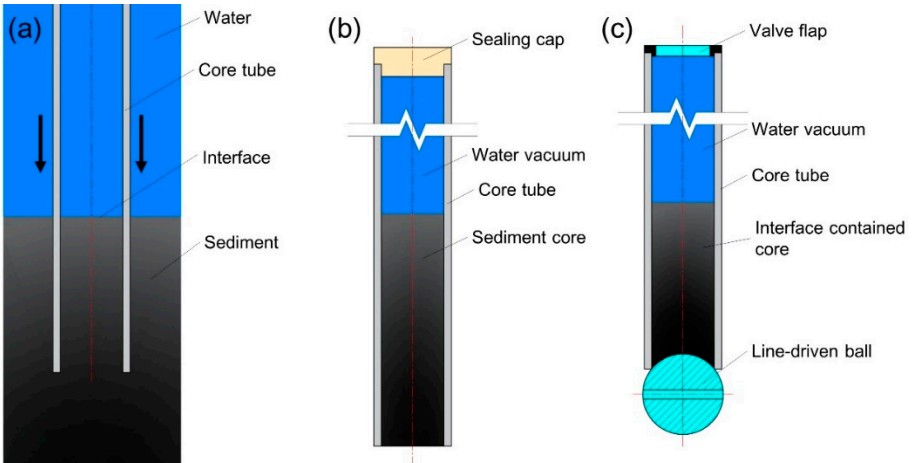

**Figure 27.** Schematic of corer without cutter: (**a**) Penetration layout, (**b**) water vacuum kept by sealing cap and sediment core, and (**c**) water vacuum kept by line-driven ball and valve flap.

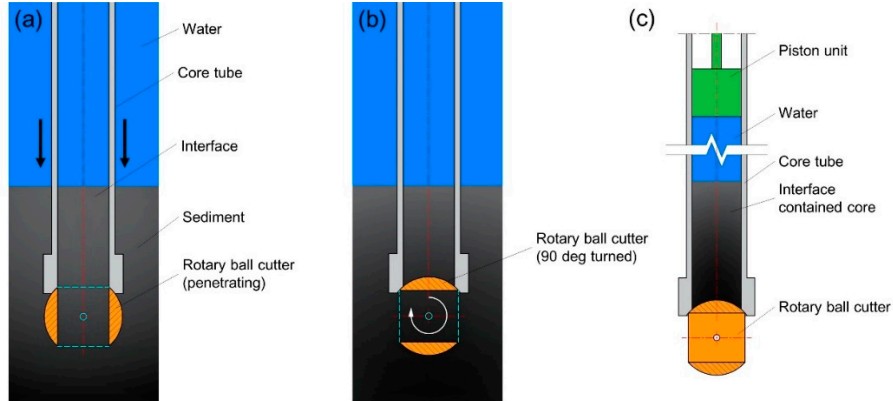

**Figure 28.** Schematic of corer with rotary ball cutter: (**a**) Penetrating, (**b**) rotary ball cutter sealing, and (**c**) core sealed.

**Table 1.** Summary of corers deployed in Antarctic subglacial aquatic sediment sampling settings.

| Year and References | Location | Corer Name & Type | Project or Unit | Core Length (cm) | Ice Thickness (m) | Water Depth (m) | Distance between Ice Surface to Sediment (m) | Coring Characteristic Items |
|---|---|---|---|---|---|---|---|---|
| 1975 [40] | Novolazarevskiy ice shelf | - | 20th Soviet Antarctic expedition | 28 | 357 | 203 | - | - |
| 1977–1978 [41] | Ross ice shelf Site J-9 | Benthos model 2171 GC | RISP | 58 cores: up to 122 | 420 | 237 | 597 | (1) Flame-jet drilling access borehole size 30–80 cm (2) 3 m long core barrel |
| 1991–1992 [8] | Fimbulsen Ice Shelf | Benthos model 2171 GC | NARE | Cores: 30 to 60 | - | 400 | - | (1) Corer total weight 110 kg. (2) 2.4 m long in corer liner with ID 67 mm |
| 1989–1993 [16,19] | Ice Stream B (UpB camp area) | Caltech PC | - | Cores: up to 400 | ~1030 | ~600 | - | (1) 6 m in corer length and liner ID 50 mm (2) 10 cm diameter borehole |
| 2000 [10] | Amery Ice shelf AM02 | Wintle GC | AMISOR | 144 | 373 | - | 843 | (1) ~40 cm diameter hot-water access borehole. (2) Purpose-built slim-line, 12 cm diameter gravity corer |
| 2003 [10] | Amery Ice shelf AM01b | | | 47 | 479 | - | 840 | |
| 2005 [10] | Amery Ice shelf AM03 | | | 60 | 722 | - | 1339 | |
| 2006 [10] | Amery Ice shelf AM04 | | | 124 | 603 | - | 1002 | |
| 2006 [35] | McMurdo ice shelf AND-1B site | AWI GC | ANDRILL | Seven WSI cores: 11 to 53 | 82 | 850 | - | (1) 80 kg weight with 5–22 m release distance (2) Equipped with plastic core tube 1.0 or 1.5 m in length and 59.5/63 mm in ID/OD (3) Open tube end |
| | | ANDRILL PushC | | Four cores: 33 to 156 | | | | (1) PQ drill string installed (2) 73 kPa shear strength at 1.94 m sediment depth |
| Before 2008 [8,9] | Hodgson Lake | UWITEC GC UWITEC KOL Kolbenlot PerC | BAS | - 3 cores up to 200 | 3.7 | - | 93.4 | (1) Surface sediments collected (2) Automatic ball valve core catcher Three 2 m cores with total penetration depth 3.76 m |

**Table 1.** *Cont.*

| Year and References | Location | Corer Name & Type | Project or Unit | Core Length (cm) | Ice Thickness (m) | Water Depth (m) | Distance between Ice Surface to Sediment (m) | Coring Characteristic Items |
|---|---|---|---|---|---|---|---|---|
| 2009 [10] | Amery Ice shelf AM05 | Wintle GC | AMISOR | 110 | 624 | - | 979 | ~40 cm diameter hot-water access borehole. |
| 2010 [10] | Amery Ice shelf AM06 | | | 88 | 607 | - | 902 | |
| 2010–2011 [38] | Ross Ice Shelf Coulman High | AWI<br><br>GC | ANDRILL | 28 cores, up to 129 | Four sites: 261 to 268 | Four sites: 798 to 864 | - | (1) Equipped with 0.5–2 m long plastic core barrel<br>(2) 5 to 20 m 'free fall' distance |
| 2011–2012 [58] | Larsen C Ice Shelf & southern GVI-IS | BAS MO-PercC | BAS | TL 1160 up to 290 | - | - | - | - |
| 2012–2013 | Pine Island Glacier PIG A. PIG B, PIG C | BAS MO-PercC | BAS | 30, 92, 115 | ~500 to 600 | WCT ~200 to 400 | - | (1) 20 cm diameter hot-water access borehole<br>(2) Type of sediment: mud, sandy mud, gravelly–sandy mud or muddy gravel<br>(3) grain size up to 8 mm |
| 2012–2013 [7,8,23,33] | Lake Ellsworth (~36 cm ID access borehole, un-accessed lake) | SLE GC | SLE | - | ~3000 | WCT ~150 | - | (1) 270 kg with 3.7 m core tube<br>(2) With cutter and core catcher |
| | | SLE BAS/UWITEC CEP-PercC | | - | | | - | (1) Real-time camera monitoring coring<br>(2) Linear motors powered hammer |
| | | SLE CEP- PushC | | - | | | - | (1) Real-time camera monitoring coring<br>(2) Push coring and ball valve sealing by single drive<br>(3) Volume compensation piston |

**Table 1.** *Cont.*

| Year and References | Location | Corer Name & Type | Project or Unit | Core Length (cm) | Ice Thickness (m) | Water Depth (m) | Distance between Ice Surface to Sediment (m) | Coring Characteristic Items |
|---|---|---|---|---|---|---|---|---|
| 2013 [1,8,63] | Lake Whillans (min 60 cm ID access borehole) | NIU/UWITEC Multi-GC | WISSARD | WSI 8 cores 20 to 40 | 797 ± 10 | 724.6 ± 0.7 | 802.6 ± 0.8 | (1) Three replicated cores can be obtained once together (2) Automatic ball valve core catcher (3) Seven unsuccessful attempts due to corer radial size |
| | | WISSARD/Caltech PC | | 80 | | | | (1) Wireline messenger trigger released (2) Guiding plates on top end |
| 2013 [1,8,63] | Lake Whillans (min 60 cm ID access borehole) | NIU CEP-PercC | WISSARD | 40 | 797 ± 10 | 724.6 ± 0.7 | 802.6 ± 0.8 | (1) Automatic ~900 kg hydraulic percussion drop mass (2) ~14 m corer length, 5 m core barrel with ID/OD 100/141 mm (3) smart winch error resulted in worked as a gravity corer |
| 2014–2015 [52] | Filchner-Ronne Ice Shelf S5A, S5B, S5C | BAS/UWITEC GC | | Seven cores: TL 265 | ~770 | WCT ~400 | - | (1) 30 cm hot water access borehole (2) Equipped with cutter and core catcher |
| | Filchner-Ronne Ice Shelf FSW1 | BAS/UWITEC GC | BAS | 33 | 853 | WCT | - | |
| 2015–2016 [52] | | BAS Mini-C | | 14 | | 471 | - | (1) 3 m long cores were expected (2) 30 cm hot water access borehole (3) GC with cutter and core catcher |
| | Filchner-Ronne Ice Shelf FSE1 | BAS/UWITEC GC | | 8 | 891 | WCT | - | |
| | | BAS Mini-C | | 73, 78 | | 528 | - | |
| | Filchner-Ronne Ice Shelf FSE2 | BAS/UWITEC GC | | 55 | 837 | WCT | - | |
| | | BAS Mini-C | | 36 | | 442 | - | |
| 2016–2017 | Filchner-Ronne Ice Shelf FNE1 | BAS Mini-C | | 64 | 615 | 440 | - | - |
| | Filchner-Ronne Ice Shelf FNE2 | BAS MO-PercC | BAS | 135.5 | 597 | 588 | - | |
| | | BAS Mini-C | | 75.5 | | | - | |
| | Filchner-Ronne Ice Shelf FNE3 | BAS MO-PercC | | 152 | 597 | 643 | - | |
| | | BAS Mini-C | | 75.5 | | | - | |

**Table 1.** *Cont.*

| Year and References | Location | Corer Name & Type | Project or Unit | Core Length (cm) | Ice Thickness (m) | Water Depth (m) | Distance between Ice Surface to Sediment (m) | Coring Characteristic Items |
|---|---|---|---|---|---|---|---|---|
| 2017–2019 (unpublished data) | Ekström Ice Shelf Neumayer Station III | AWI GC | AWI | WSI cores: up to ~46 | 187 to 332 | 118 to 656 | ~270 to 700 | (1) un-functioned valve caused more than half GC runs without core (2) 5 to 15 m release distance (3) Core tube end broken in some runs |
| 2017–2019 (unpublished data) | Ekström Ice Shelf Neumayer Station III | BAS/UWITEC MO-PercC | AWI | Cores: 32 to 200 | 187 to 332 | 118 to 656 | ~270 to 700 | (1) Corer max OD 140 mm, 3 m core barrel (2) 3 hammer weights together ~33 kg |
| | | JLU VC | | - | | | | (1) 270 mm max OD, 256 kg, 50 Hz self-synchronous vibro-head (2) Self-contained batteries with time-delay power-on |
| | | JLU VC+ | | - | | | | (1) Cut core barrel cross-section to 50% (2) Time-delay power-on with intermittent vibration mode (3) Standard Makita batteries |
| | | JLU MO-HC | | WSI 128 | | | | (1) Tube only with cutter, core catcher and ball valve (2) Long core with well-preserved water–sediment interface (3) 38 kg hammer |
| 2018–2019 | Rutford Ice Stream | BAS MO-PercC | BAS | ~10 | 2152 | - | - | The deepest hot water drilled subglacial access borehole |
| 2018–2019 [56,57] | Lake Mercer | NIU/UWITEC Multi-GC | SALSA | Six cores with WSI | - | 15 | 1084 | (1) ~60 cm diameter access borehole (2) At least 6 cores with water–sediment interface obtained 9.1 m in corer length and 6 m core barrel |
| | | WHOI GC | | 70, 170 | | | | |

GC—Gravity Corer, PC Piston Corer, HC—Hammer Corer, PerC—Percussion Corer, PushC—Push Corer, VC—Vibrocorer; CEP—Cable Electric Powered, MO—manually operated; AMISOR—Amery Ice Shelf Oceanographic Research, ANDRILL—Antarctic geological drilling, AWI—Alfred Wegener Institute, BAS—British Antarctic Survey, JLU—Jilin University (China), NARE—Norwegian Antarctic Research Expedition, NIU—Northern Illinois University, SALSA—Subglacial Antarctic Lakes Scientific Access, SLE—Subglacial Lake Ellsworth, SLW—Subglacial Lake Whillans, UWITEC—an established limnological engineering company based in Austria (http://www.uwitec.at), WHOI—Woods Hole Research Institute; ID—inner diameter, OD—outer diameter, TL—Total Length, WSI—Water–sediment Interface (obtained), WCT—water column thickness.

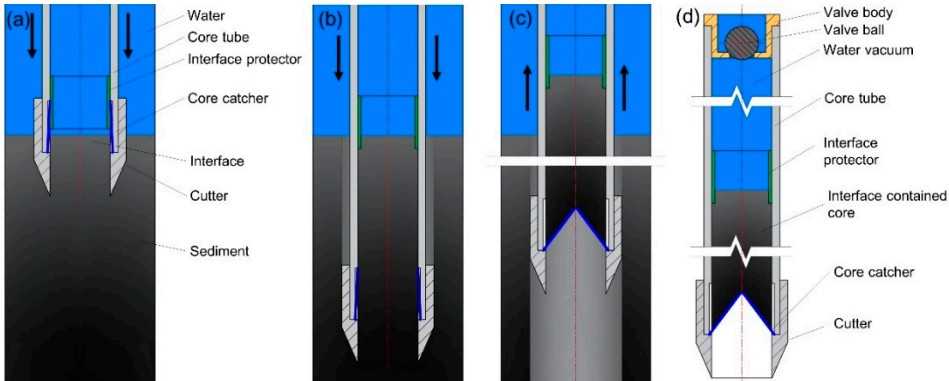

**Figure 29.** Schematic of corer with interface protector: (**a**) The cutter touches to the sediment, (**b**) penetration, (**c**) pulling out from sediments, and (**d**) core sealed.

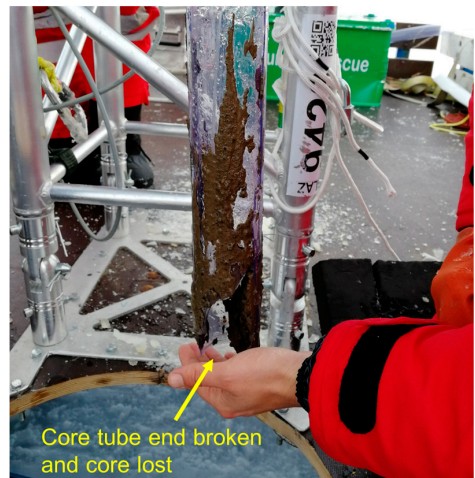

**Figure 30.** Broken tube end of gravity corer after meeting with hard material in sediments (photograph was taken by Y. Li).

### 3.1.2. Coring Methods for Obtaining Long Cores

If an undisturbed water–sediment interface is not required, coring methods can be much more robust and incorporate additional energetic devices to enable deeper penetration. Since the access borehole imposes a size limit, most lake/ocean sampling devices or techniques are inapplicable (e.g., trigger system, stabilizing frame, more powerful vibro-head). However, there are still several specific power unit designs that can help the core barrel go deeper into sediments. These power units have a cylindrical shape, designed to cope with radial size limitation caused by the access borehole diameter. Corers for obtaining long sediment cores were equipped with increased potential energy (e.g., heavier gravity corer), continuous kinetic energy input (e.g., hammer/percussion corer), or a penetration resistance reducing mechanism (e.g., vibrocorer).

(1) Gravity corers with increased potential energy;

Without the help of a 'bottom touch' trigger system, the gravity corer must be released several meters above seafloor by the winch on ice surface. As the corer's 'free fall' speed reaches a constant speed with no further increase [70], the simplest and most effective method to increase the penetration energy of a gravity corer is to increase its body weight, especially when the corer diameter and core tube size is limited. For example, the SLE gravity corer, whose core tube section area is similar to the UWITEC gravity corer, was designed with a 270 kg head weight and a 3.7 m lined core barrel with a metal cutting head [8]. It should be noted that the 170 cm long subglacial lake sediment core record was achieved by the WHOI heavy gravity corer, which had a total length of 9.1 m.

(2)　Corers with cylindrical hammer/percussion unit;

To deal with the size limit imposed by the access borehole, corers with a hammer or percussion unit installed in the upper core barrel end have been developed. The hammer or percussion head, manually operated or powered by an electric cable, is cylindrical to permit the entire corer assembly to pass through the access borehole.

The BAS/UWITEC manually operated percussion corer can accomplish all operations by a single cable [46]. Its ultra-compressed design enables the corer to achieve a maximum diameter of only 14 cm and, thus, can applied in most hot water boreholes. The longest 290 cm long sub-ice-shelf sediment core was obtained by this corer at the Larsen C and George VI ice shelves [13,58].

Another manually operated hammer corer is the JLU hammer corer, which was built for sediment sampling within the 2018–2019 Sub-EIS-Obs project. This corer shares a similar hammer operation process with the BAS/UWITEC manually operated percussion corer, but differs in its core tube structure and core retaining mechanism. Based on published data, the 128-cm-long core and its well-preserved water–sediment interface, is the longest undisturbed core ever obtained by cable-suspended tools from Antarctic subglacial sediments.

Using high-strength fiber as the conducting cable, two other advanced percussion corers have been recently developed, the NIU percussion corer and the SLE percussion corer. The NIU percussion corer contains a hydraulic motor that can drive a ~900 kg percussion drive mass and then hammer a 5 m long core barrel into stiff over-consolidated sediment [33]. The SLE percussion piston corer, designed with a pressure chamber, contains a hammer unit driven by a liner motor. A real-time camera installed within the corer helps the surface personnel to operate the corer precisely [7,8]. Unfortunately, both electric cable-powered corers have failed to obtain cores in Antarctica due to reasons that are not related to the corers' design or performance.

(3)　Corer with cylindrical vibro-head;

Vibrocoring is a technique commonly used to collect cores in any type of sediment. The vibrocorer is equipped with a vibro-head at the upper end of the core barrel and utilizes the sediment 'vibro-liquefaction' phenomenon to increase its penetration depth [65]. Vibrocoring techniques have been questioned because of the core disturbance caused by the vibrations, but recent research results show no evident core differences between vibrocoring and gravity coring [68]. However, there are two technical challenges in subglacial sediment coring with vibrocoring, including how to pass the vibro-head through the access borehole and how to switch the vibration on/off after the corer is underneath the ice. JLU has proposed two designs for vibrators (Figure 31). One is powered and controlled by an electric armored cable and the other is powered and controlled by downhole batteries and time-relays. The two vibro-motors in each vibrator can reach synchronous states without any mechanical transmission. However, the time-relay delayed on/off switch is unwanted in field operations because the operational time is difficult to control.

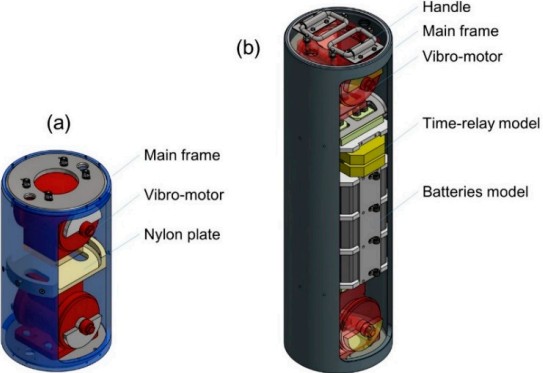

**Figure 31.** Two versions of JLU cylindrical vibrator: (**a**) Powered through electric cable and (**b**) powered by self-contained batteries.

### 3.2. Discussions

Results of coring aiming to obtain the water–sediment interface are shown in Figure 32. Except for the 128 cm long core recovered by a hammer corer, all other cores are shorter than 60 cm. The main reason for this is that the tube end without a core catcher protection is unable to cope with gravel. The corer should be designed as light as possible. Otherwise, 'free-falling' from several meters above the seafloor will damage the tube end, as shown in Figure 30.

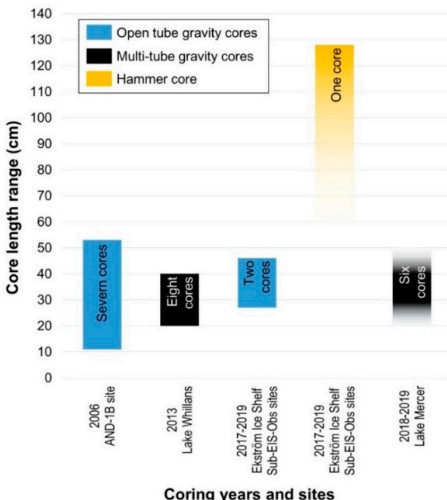

**Figure 32.** Coring results aimed to recover undisturbed sediment/water interface.

The results of coring that aimed to obtain long cores are shown in Table 1 and Figure 33. The most popular coring tools are gravity corers, which are also the most cost-effective and simplest in design, construction, and deployment. However, the lengths of all cores obtained with gravity corers have been shorter than 170 cm. The longest 170 cm long core was recently recovered from subglacial Lake Mercer in the 2018–2019 season using the WHOI corer [55]. The Caltech piston corer holds the record of 400 cm for a core recovered from a sub-ice-stream setting [19], however, information about this corer is very limited. There are three electric-powered coring tools currently developed for long cores. Unfortunately, all of these have failed to reach their intended target. Finally, manually operated percussion corers are the most feasible and economically applicable method to obtain long sediment cores from subglacial aquatic settings. These corers hold the record for the longest (290 cm) sub-ice-shelf sediment core [13] using cable-suspended coring methods.

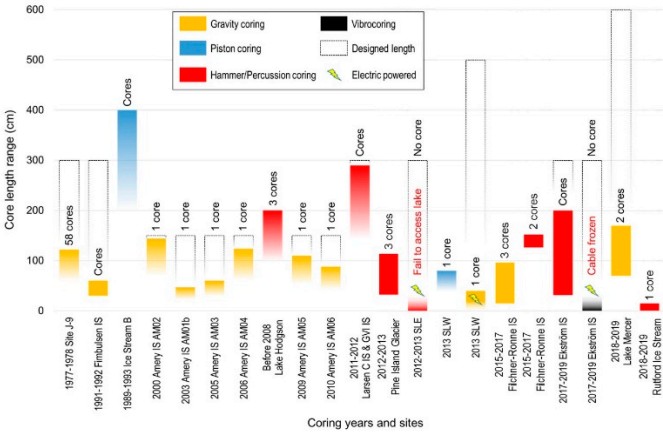

**Figure 33.** Coring results aimed to recover long size cores (IS—ice shelf, GVI—George VI, SLE—Subglacial Lake Ellsworth, SLW—Subglacial Lake Whillans).

## 4. Future Prospects

### 4.1. Methods to Increase Penetration Depth

(1)   High performance power unit

Measures such as the use of a heavier dead weight, a more powerful hammer unit, and higher frequency and amplitude vibrations, have already been proven to increase core penetration in ocean and lake sediment coring [71,72]. This has been a challenging task because the radial size, the cable or battery capacity, and the winch lifting ability are all strictly limited. Another options to increase penetrating power is through additional actuators. If this is implemented, then a real-time control system at the ice surface and a more powerful cable may be required. Another idea is to use a multi-dynamic power head, for example, a power head that contains both a hammer and vibrators. Furthermore, it is possible to build a hydraulic vibrocorer or a hammer corer that is powered by a hot water hose or a hydrostatic energy accumulator (Figure 34). The technological superiority of the hydraulic corers (e.g., the Selcore corer) has already been proven in ocean sediment coring. These can work in water depths of up to 3500 m and can penetrate at depths that are two times more than that achieved by the gravity corer of an equivalent weight [71,73].

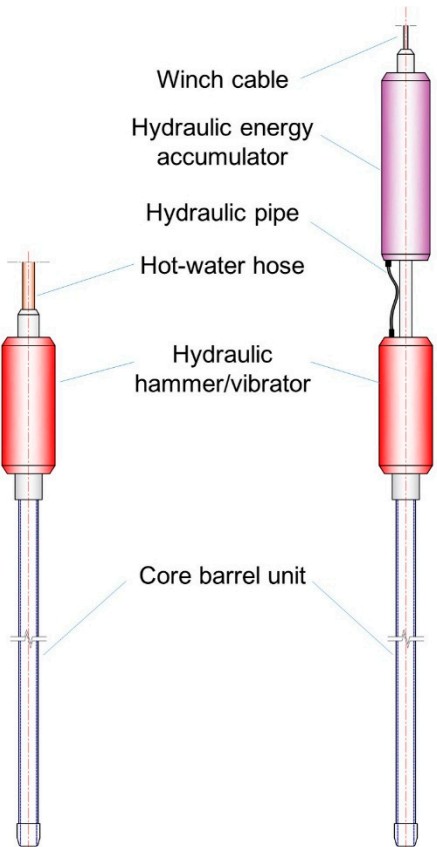

**Figure 34.** Schematic of hydraulic powered corers.

(2)   Cable-suspended rotation drilling

Cable-suspended rotation sediment drilling tools are not easy to adopt for subglacial aquatic settings because there is not enough counter torque to offset the torque produced by the drill head. One solution is to use a DBTS system as the power head, which will reduce the torque. In this case, the core barrel should contain a static inner barrel (Figure 35); otherwise, the core will be totally twisted and unsuitable for analysis.

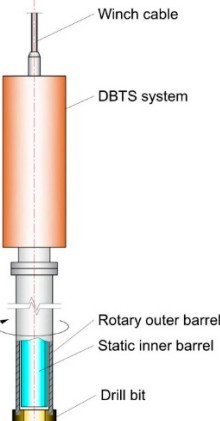

**Figure 35.** General layout of cable suspending sediment corer based on the DBTS system.

(3)    Reinforced cutter material

The tube end and/or the cutter's circular edge are the most vulnerable areas during penetration, especially when the corer meets hard material, such as gravel (Figures 30 and 36). For the open tube end corer, a metal cutter can be useful to prevent splitting. For the metal cutter, a stiffer material (e.g., 40Cr13 steel [74]) and high-strength surface coating (e.g., gas detonation [75,76]) can provide a longer working lifetime.

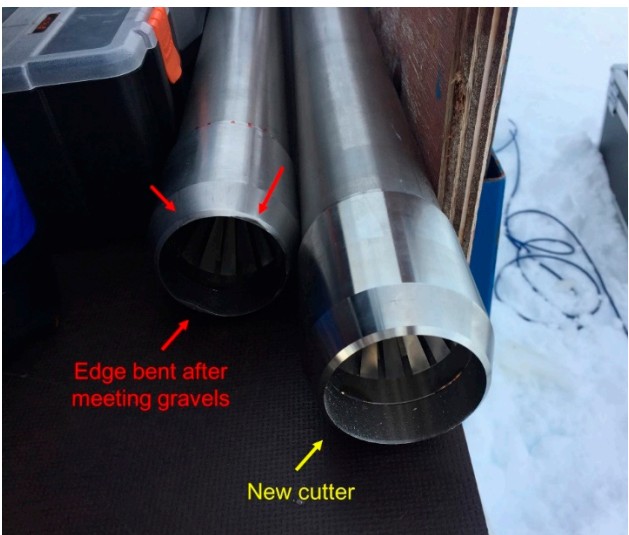

**Figure 36.** Photo comparison between the new cutter and the edge bended cutter after meeting gravel (photograph was taken by D. Gong).

(4)    Hydrophobic surface coating

Viscosity and friction between sediments and the core tube's inner/outer surfaces are the main factors that prevent the core tube from penetrating deeper. They can compress the core and induce a 'plugging effect' [77]. One solution is to use a hydrophobic coating on the cutter, core barrel, liner, and even the core catcher blades [78,79].

*4.2. Continuous Coring at the Same Position*

The UWITEC company developed a continuous coring method from a platform floating at the lake surface. In this method, the corer can repeatedly sample through the same sediment hole. It is

guided by two tethers linked between the floating platform and the upturned cone-shape platform. The upturned cone-shape platform is lowered to the lake bottom by the two tethers, and its bottom tube works as a casing to prevent the soft surface sediments from falling through the borehole. However, this economically convenient and high-yielding technique cannot be used due to two reasons, the borehole limits deployment on guiding platform and the guiding tethers are apt to freeze.

We suggest that continuous cable-suspended coring in subglacial aquatic sediments (Figure 37) be based on the UWITEC cable-suspended continuous coring method. The platform is cylindrical and can pass through the access borehole and penetrate the sediment. The lead ropes link between the surface winch and the underwater platform and the cable-suspended corer can repeatedly penetrate the same sediment borehole guided by the lead ropes. Though, hole refreezing and freezing in of the lead ropes, particularly in deep holes, remains a likely problem.

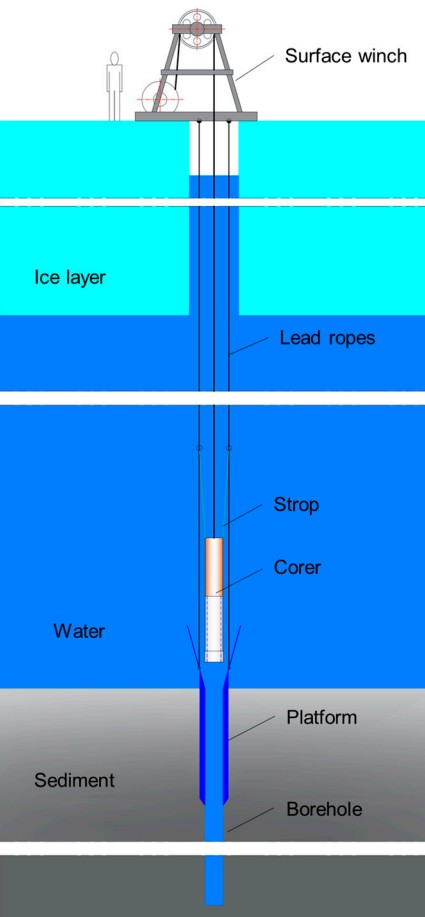

**Figure 37.** Schematic of subglacial aquatic sediment continuous coring.

### 4.3. Long Core with Undisturbed Water–Sediment Interface

It is challenging to obtain long cores with an undisturbed water–sediment interface. The reasons are as follows: (i) To obtain a long core, a stiff, orange peel-shaped core catcher needs to prevent the core from being lost during lifting; however, such a core catcher totally destroys the soft surface sediments and (ii) the interface passes through the whole core length and the soft material is disturbed due to inner side friction during this process.

The water–sediment interface protection technique developed by JLU can be a method to attain two targets in one run. The advantages of this technique are the following: (i) It combines two corer deployment times into one and provides the opportunity to deploy other devices into the access borehole and (ii) it can be used in all corer types with a cutter and orange peel type core catcher.

However, the surface operation for such cores should be re-considered, especially on double-tube core barrel unit structures.

**Author Contributions:** This review paper was finished by D.G., X.F., Y.L., B.L., N.Z., R.G., E.C.S., W.D., S.B., O.E., J.T., B.K.B., N.K., F.W., B.B., Y.L., Y.Y., X.L., A.L., and P.T. The statements of author contributions are as following: conceptualization, D.G. and P.T.; methodology, D.G., X.F., and P.T.; software, D.G. and B.L.; validation, D.G., X.F., F.W., and P.G.; formal analysis, D.G., Y.Y., and P.T.; investigation, D.G., X.F., Y.L., B.L., N.Z., Y.L., X.L., A.L., and P.T.; resources, D.G., X.F., R.G., E.C.S., W.D., S.B., O.E., J.T., B.K.B., N.K., F.W., B.B., and P.G.; data curation, D.G., Y.Y., X.L., A.L., and P.T.; writing—original draft preparation, D.G. and P.T.; writing—review and editing, D.G. and P.T.; visualization, D.G. and P.T.; project administration, P.T.

**Funding:** This research was funded by the National Science Foundation of China (grant number 41327804) and the Program for Jilin University Science and Technology Innovative Research Team (grant number 2017TD-24).

**Acknowledgments:** The authors thank AWI for logistical and financial support of field operations with JLU corers in Antarctica in 2017–2018 and 2018–2019 seasons within Sub-EIS-Obs project. We thank Richard Niederreiter for providing data of UWITEC corers. The authors also express their gratitude to all teachers, engineers, and postgraduate students of the Polar Research Center at Jilin University for their help in designing and testing JLU corers. We thank Bowen Liu and Han Zhang for their help in corer laboratory testing.

**Conflicts of Interest:** The authors declare no conflict of interest and approve this version of the paper.

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
