# Peer review of "Coring of Antarctic Subglacial Sediments"

_jmse, doi:10.3390/jmse7060194_

Reviewer 1 Report

Please see the attached review document.

Author Response

Detail response for Review report 1

Review of Coring of Antarctic subglacial sediments by Da Gong et al., J. Mar. Sci. Eng.

General comment:

This manuscript reviews and describes the various sediment coring technologies used in obtaining subglacial sediments cores, via ice boreholes, in aquatic environments where liquid water is present in the soft sediments. ‘Aquatic’ needs to be emphasized at the earliest opportunity in the manuscript. There is a summary of the corers deployed and the cores recovered in Antarctica. The manuscript structure is logical and well set out; the text is overall well written and complimented by clear and concise figures with supporting photographs.

I would recommend only minor corrections are required prior to publication.

A small number of core recovery sites have been overlooked and further details are provided at the end of this review.

Minor comments and suggested changes:

Comments and Suggestions No.1L18 change to: ‘subglacial aquatic environments’. This makes it clear from the outset that this review is only concerned with water-saturated sediments and not sediments, which are frozen in a sediment-ice matrix

AnswerRevised

Replaced ‘subglacial environments’ with ‘subglacial aquatic environments’.

Comments and Suggestions No.2L20 change to: even earlier than ice cores,

AnswerRevised

Changed ‘even earlier than the ice core’ to ‘even earlier than ice cores’.

Comments and Suggestions No.3L20 delete: which are restricted to the age of the ice itself

AnswerRevised

Deleted sentence ‘which are restricted to the age of the ice itself’.

Comments and Suggestions No.4L25 replace: ‘and then touch’ with ‘to reach’

AnswerRevised

Replaced ‘and then touch’ with ‘to reach’.

Comments and Suggestions No.5L26 delete: ’layers’

AnswerRevised

Deleted ‘layers’ from sentence ‘through ice layers up to 3000–4000 m thick’.

Comments and Suggestions No.6L29 change to: ‘pass through access boreholes and collect the sediment core.’

AnswerRevised

Changed ‘pass through the access boreholes and then collect the sediment core.’ to ‘pass through access boreholes and collect the sediment core.’

Comments and Suggestions No.7L39 is it necessary to mention hard drilling strings here? I would suggest just focusing on the limitations presented by the environment. Perhaps just delete: ‘and no continuous coring method has ever been developed for coring through ice more than 100 m thick, with the exception of the string deep drilling techniques.’

AnswerRevised

Deleted sentence ‘and no continuous coring method has ever been developed for coring through ice more than 100 m thick, with the exception of the string deep drilling techniques.’

Comments and Suggestions No.8L41 delete ‘special’

AnswerRevised

Deleted ‘special’ from sentence ‘the special sedimentary structure and the coring techniques.’

Comments and Suggestions No.9L43 change to ‘the working principles, corer characteristics, operational’

AnswerRevised

Changed sentence ‘the working principles, characteristics, operational’ to ‘the working principles, corer characteristics, operational’

Comments and Suggestions No.10L50 suggested change: ‘Subglacial sediments in aquatic environments are found at the seabed beneath ice shelves, subglacial lake beds, and at the base of ice streams.

AnswerRevised

Changed ‘Subglacial sediments occur in aquatic environments such as sub-ice-shelves, sub-ice-streams, and subglacial lakes.’ to ‘Subglacial sediments in aquatic environments are found at the seabed beneath ice shelves, subglacial lake beds, and at the base of ice streams.’

Comments and Suggestions No.11L54 and elsewhere. Delete ‘etc’. Either complete the list of things to mention or use ‘for example’.

AnswerRevised

Deleted all ‘etc’ in the paper content or replaced ‘etc’ by ‘for example’.

Comments and Suggestions No.12Figure 1-

Change ‘Basal ice flow’ to ‘Basal ice’ or ‘Accretion ice’

Change the central figure panel to an ice stream by removing the lake.

Only have subglacial water flow arrows at the ice-sediment interface

AnswerRevised

See Figure 1

Changed ‘Basal ice flow’ to ‘Basal ice’.

Changed the central figure panel to ice stream, removed the lake.

Deleted other flow narrows, only left the flow narrows at the ice-sediment interface.

Comments and Suggestions No.13L65 change: ‘explored’ to ‘characterised’

AnswerRevised

Changed ‘explored’ to ‘characterised’.

Comments and Suggestions No.14L67 change ‘matched bathymetries’ to ‘determining sub-ice bathymetry’

AnswerRevised

Changed ‘matched bathymetries’ to ‘determining sub-ice bathymetry’

Comments and Suggestions No.15L73 change ‘such as’ to ‘from

AnswerRevised

Changed ‘such as’ to ‘from’.

Comments and Suggestions No.16L74 change to: overlying ice, hydrological flow [12],

AnswerRevised

Changed ‘overlying ice bottom, hydrological motion’ to ‘overlying ice, hydrological flow’

Comments and Suggestions No.17L75 change ‘derived’ to ‘deposited’

AnswerRevised

Changed ‘derived’ to ‘deposited’.

Comments and Suggestions No.18L75 change to: ‘Sediments beneath ice shelves are most likely derived from two main sources: the open marine environment and basal debris from the ice shelf base near the grounding zone [10].

AnswerRevised

Changed ‘Sediments beneath ice shelves are most likely derived from two main sources: the ocean current and ice shelf base debris formed at the transition area from the grounding zone to the borehole site [10].’ to ‘Sediments beneath ice shelves are most likely deposited from two main sources: the open marine environment and basal debris from the ice shelf base near the grounding zone [10].’

Comments and Suggestions No.19L82-83 charge to ‘mainly on rheology and its important to ice streaming and the mass

AnswerRevised

Changed ‘mainly on mechanics and sedimentary record of the ice streams, and the important influence of ice streaming on the mass’ to ‘mainly on rheology and its important to ice streaming and the mass’.

Comments and Suggestions No.20L85 change to: ‘to determine when the ice sheet last decayed,’

AnswerRevised

Changed ‘to determine the last decay date of the ice sheet’ to ‘to determine when the ice sheet last decayed’

Comments and Suggestions No.21L92-93 change to: ‘efforts are focused on: i) what form of microbial life exists in Antarctic subglacial lakes [1] [9] [20] [21]; ’

AnswerRevised

Changed ‘including: i) what form of microbial life exists in Antarctic subglacial lakes [1] [21];’ to ‘efforts are focused on: i) what form of microbial life exists in Antarctic subglacial lakes [1] [9] [20] [21]; ’

Comments and Suggestions No.22L98 change to: ‘Based on these’

AnswerRevised

Changed ‘Based on the’ to ‘Based on these’.

Comments and Suggestions No.23L101 delete ‘-long-cylinder’

AnswerRevised

Deleted ‘long-cylinder-shaped’.

Comments and Suggestions No.24L105 delete ‘pre-‘

AnswerRevised

Deleted ‘pre-’.

Comments and Suggestions No.25L107-108 change to: ‘When the hot-water drill penetrates the ice base, the borehole water level will adjust to the local hydrological level, achieving pressure equilibrium [3].

AnswerRevised

Changed ‘After the hot-water drill reaches water reservoir, water will fill the borehole up to a certain height to achieve pressure equilibrium [3].’ to ‘When the hot-water drill penetrates the ice base, the borehole water level will adjust to the local hydrological level, achieving pressure equilibrium [3].’

Comments and Suggestions No.26L108-L111 change to: ‘The sediment coring tools are then deployed first through the air filled and then water filled parts of the borehole, through the ocean or lake water column if present, to reach the sediment, which is sampled and retrieved to the surface as sediment core samples [8].’

AnswerRevised

Changed ‘The sediment coring tools are through the borehole and into the subglacial water, then reaching the water floor and finally the sediment, which is sampled and retrieved to the surface as sediment core samples [8].’ to ‘The sediment coring tools are then deployed first through the air filled and then water filled parts of the borehole, through the ocean or lake water column if present, to reach the sediment, which is sampled and retrieved to the surface as sediment core samples [8].’.

Comments and Suggestions No.27L113 change to: ‘rate that depends on the ice temperature surrounding the borehole, water pressure, initial borehole size, water composition

AnswerRevised

Changed ‘rate that depends on borehole temperature, water pressure, initial borehole size, water composition’ to ‘rate that depends on the ice temperature surrounding the borehole, water pressure, initial borehole size, water composition’.

Comments and Suggestions No.28L116 change ‘be’ to ‘become’

AnswerRevised

Changed ‘be’ to ‘become’.

Comments and Suggestions No.29L119 change ‘applicable’ to ‘appropriate

AnswerRevised

Changed ‘applicable’ to ‘appreciate’.

Comments and Suggestions No.30L128-133 change to: ‘The ANDRILL (ANtarctic geological DRILLing) project on Ross Ice Shelf [32] used a basic winch and tower, high-strength cable (with or without power transmission capacity), a control unit, and a platform for sub-ice sediment coring. The cable, winch and tower were designed to manage loads from between 1500 to 4500 kg [33]. The operational height between the tower top pulley to the

AnswerRevised

Changed ‘Although the logistics and cost involved in the ANDRILL (ANtarctic geological DRILLing) project in the Ross Sea [32] has been great, most sub-ice sediment coring is performed with only the basic surface cable suspension facilities such as a simplified winch and tower, high-strength cable (with or without power transmission capacity), a control unit, and a platform. The cable, winch and tower are always interactively designed to manage loads from between 1500 to 4500 kg [33]. The operational height between the top pulley to the’ to ‘The ANDRILL (ANtarctic geological DRILLing) project on Ross Ice Shelf [32] used a basic winch and tower, high-strength cable (with or without power transmission capacity), a control unit, and a platform for sub-ice sediment coring. The cable, winch and tower were designed to manage loads from between 1500 to 4500 kg [33]. The operational height between the tower top pulley to the’.

Comments and Suggestions No.31L143 delete ‘layers’ as it sounds as if the layers are 4-64 mm thick.

AnswerRevised

Deleted ‘layers’.

Comments and Suggestions No.32L145 delete ‘towards a target’

AnswerRevised

Deleted ‘towards a target’.

Comments and Suggestions No.33L150 change ‘structures’ to ‘structure’

AnswerRevised

Changed ‘structures’ to ‘structure’.

Comments and Suggestions No.34L156 Delete ‘Aside from the major deep drilling efforts in the Ross Sea’ as the major effort was associated with the deep rock coring and not the sediment coring.

AnswerRevised

Deleted ‘Aside from the major deep drilling efforts in the Ross Sea’.

Comments and Suggestions No.35Figure 3

Labels for the Filchner-Ronne sites need to be swapped.

Additional sites can be added on Filchner-Ronne, Pine Island Glacier, and Rutford Ice Stream (details at end of review)

AnswerRevised

Swapped labels for the Filchner-Ronne sites.

Added sites of Filchner-Ronne, Pine Island Glacier, and Rutford Ice Stream based on the end review.

See updated figure 3.

Comments and Suggestions No.36L187 add ‘(Fig 2b)’ at end of line.

AnswerRevised

Added ‘(Fig 2b)’ at end of sentence ‘Due to the borehole size restriction, the radial the trigger system must be removed.’.

Comments and Suggestions No.37L201 change ‘in’ to ‘on’ or ‘beneath

AnswerRevised

Revised ‘in’ to ‘on’.

Comments and Suggestions No.38L203 delete ‘in Australia’

AnswerRevised

Deleted ‘in Australia’.

Comments and Suggestions No.39L205-07 change to: ‘thickness was about 420 m, marine water column was 237 m and’

AnswerRevised

Changed ‘thickness was near 420 m, marine water column measured 237 m and’ to ‘thickness was about 420 m, marine water column was 237 m and’.

Comments and Suggestions No.40L214 change ‘applied’ to ‘used’

AnswerRevised

Changed ‘applied’ to ‘used’.

Comments and Suggestions No.41L216 change ‘edge’ to ‘front’

AnswerRevised

Changed ‘edge’ to ‘front’.

Comments and Suggestions No.42L230 change to: ‘that was a UWITEC standard plastic liner, 1.0 or 1.5 m long and 59.5/63 mm in ID/OD’

AnswerRevised

Changed ‘that is actually a UWITEC standard plastic liner that is 1.0 or 1.5 m long and 59.5/63 mm in ID/OD’ to ‘that was a UWITEC standard plastic liner, 1.0 or 1.5 m long and 59.5/63 mm in ID/OD’.

Comments and Suggestions No.43L234 delete ‘ or a force load

AnswerRevised

Deleted ‘or a force load’.

Comments and Suggestions No.44239 delete ‘pressure

AnswerRevised

Deleted ‘pressure’

Comments and Suggestions No.45L250 change ‘within’ to ‘with’

AnswerRevised

Changed ‘within’ to ‘with’.

Comments and Suggestions No.46L273 change ‘common’ to ‘standard’

AnswerRevised

Changed ‘common’ to ‘standard’.

Comments and Suggestions No.47L300 change to ‘through a tether

AnswerRevised

Changed ‘through tether’ to ‘through a tether’.

Comments and Suggestions No.48L314 change to: ‘gravity corer has not been used.’

AnswerRevised

Changed ‘gravity corer was not been used.’ to: ‘gravity corer has not been used.’.

Comments and Suggestions No.49L334 change to: ‘liner to retain the sediment core.’

AnswerRevised

Changed ‘liner to keep the sediment core.’ to ‘liner to retain the sediment core.’.

Comments and Suggestions No.50L342 change ‘added weights’ to ‘, making

AnswerRevised

Changed ‘added weights’ to ‘making’.

Comments and Suggestions No.51L353-358 consider adding further details of the BAS/UWITEC corer use. (Details at the end of this review)

AnswerRevised

Added details of the BAS/UWITEC gravity/mini corer application in ‘2.1.5. BAS/UWITEC gravity corer’ as following:

‘…in the 2015–2016 field season [53]. Three cores with length up to 75.5 cm were obtained from BAS drill sites on Filchner-Ronne Ice Shelf during 2016–2017 field season, where ice thickness ranges between 597–615 m and water column thickness ranges between 440–643 m.’ (other details are updated in Table 1)

Added details of the BAS/UWITEC percussion/main corer application in ‘2.1.5. BAS/UWITEC gravity corer’ as following:

‘…with a maximum penetration of 290 cm [14] [58] [59]. BAS recovered several cores used the BAS/UWITEC corer, include: i) cores with length up to 115 cm from beneath Pine Island Glacier during 2012–2013; ii) 135.5 cm long core from FNE2 drilling site during 2016–2017; iii) 152 cm long core from FNE3 drilling site during 2016–2017; iv) ~10 cm long coarse gravel core from beneath Rutford Ice Stream during 2018–2019 (ice thickness 2152 m, the deepest hot water drilled access borehole), respectively. This corer was deployed at…’(other details are updated in Table 1)

Comments and Suggestions No.52L366 change ‘designed and built’ to ‘used’

AnswerRevised

Change ‘designed and built’ to ‘used’.

Comments and Suggestions No.53L373 when the additional step is mentioned, does this refer to the wide housing on the core barrel being used as a stop against the surface sediments whilst preserving the surface inside the core barrel from over penetration. If so, please expand the explanation and indicate on figure 14.

AnswerRevised

Added one sentence to explain function of the ‘additional step’, as following: ‘The coring procedure is generally the same as that for the BAS/UWITEC, except for an additional step included to preserve the water-sediment interface. The water-sediment interface can be kept with an undisturbed state in the liner, even after the corer lifted to ice surface.’

Indicated ‘Core catcher and water-sediment interface protector’ in Figure 14.

Working principle, structure and steps are shown in ‘3.1.1. Coring methods for obtaining un-disturbed water-sediment interface’ and ‘Figure 29’

Comments and Suggestions No.54L387 what is the maximum diameter of this NIU corer?

AnswerRevised

Maximum diameter of this NIU corer is ~220 mm (8.625 inch).

Added ‘~220mm maximum OD’ into the sentence and now it is ‘The corer is designed to be sterilized and it measures a total length of ~14 m, ~220mm maximum OD, including a 5-m long core barrel with ~100/141 mm ID/OD.’

Comments and Suggestions No.55L481-482 change to: ‘automatically initiates the sealing phase, i.e. the lead screw continues to rotate, moving the ball drive plate down, which rotates the ball valve by 90° and seals the core tube end [21]

AnswerRevised

Changed ‘automatically initiates the sealing phase, i.e. the core tube stop moves deeper, allowing the ball valve to rotate 90° to seal the core tube end [21]’ to ‘automatically initiates the sealing phase, i.e. the lead screw continues to rotate, moving the ball drive plate down, which rotates the ball valve by 90° and seals the core tube end [21]’.

Comments and Suggestions No.56L487 change ‘after’ to ‘when’

AnswerRevised

Changed ‘after’ to ‘when’

Comments and Suggestions No.57L546 change to: ‘it fell on its side at the seafloor,’ (if this is what happened?)

AnswerRevised

Yes, this correction is correct. Changed ‘it fell’ to ‘it fell on its side at the seafloor’

Comments and Suggestions No.58L551 what is the barrel diameter of the two barrels?

AnswerRevised

The custom-made core barrel ID/OD is 106/124 mm with liner inside, and the conventional UWITEC core barrel ID/OD is 59.5/70 mm.

Changed to ‘(2) the custom-made core barrel (ID/OD is 106/124 mm with liner inside) was replaced with the conventional UWITEC core barrel (ID/OD is 59.5/70 mm with liner inside), thus reducing the cutting area by half;’

Comments and Suggestions No.59L551 change ‘guider’ to ‘liner

AnswerRevised

Dear reviewer, this ‘guider’ is actually a stainless steel block which was designed to prevent core barrel falling down, therefore, better to keep it as ‘guider’.

Comments and Suggestions No.60L565 change to: ‘designed to be sterilized for sediment coring

AnswerRevised

Changed ‘designed for sterilized sediment coring’ to ‘designed to be sterilized for sediment coring’.

Comments and Suggestions No.61L569 change ‘and’ to ’with

AnswerRevised

Changed ‘and’ to ’with’.

Comments and Suggestions No.62L570 should ‘cathead’ be ‘capstan’?

AnswerRevised

Yes, that is correct. Changed ‘cathead’ to ‘capstan’.

Comments and Suggestions No.63L571 change ‘starts to work in case of’ to ‘operate to mitigate against the’

AnswerRevised

Changed ‘starts to work in case of’ to ‘operate to mitigate against the’.

Comments and Suggestions No.64L594 Change to ~400 cm longs cores from Up-B site

AnswerRevised

Changed ‘170-cm long core retrieved from the bottom of subglacial Lake Mercer’ to ‘~400 cm longs cores from Up-B site’

Comments and Suggestions No.65Table 1

Filchner-Ronne sites – reference is not [54] – 2014-15 ice thickness ~770 m and wct ~400 m. Consider adding other BAS/UWITEC corer sites (list at end of review)

AnswerRevised

In Table 1:

Deleted reference label [54] about Filchner-Ronne sites.

Revised ice thickness of 2014-2015 Filchner-Ronne Ice Shelf drilling site from ‘750~891’ to ‘770’.

Added WCT water column thickness from blank to ‘WTC ~400’. 

Comments and Suggestions No.66L654 delete ‘set’

AnswerRevised

Delete ‘set’.

Comments and Suggestions No.67L660 how heavy is the WHOI corer?

AnswerRevised

The weight of the WHOI gravity corer is about 2500 pound (~1130 kg). We added it in to the first paragraph of ‘2.1.7. WHOI gravity corer’, as following: ‘The WHOI (Woods Hole Research Institute) gravity corer (Fig. 11) is the largest tool in SALSA’s lineup, with a total length of 9.1 m, its purpose is to take up to ~6.0 m core samples by its heavy weight (~1130 kg).’.

Comments and Suggestions No.68L728 change ‘option’ to ‘options

AnswerRevised

Changed ‘option’ to ‘options’.

Comments and Suggestions No.69L729 delete brackets

AnswerRevised

Deleted ‘(e.g., weight, hammer energy, vibrator)’

Comments and Suggestions No.70Figure 36

Change label to ‘Edge bent after meeting gravel’

AnswerRevised

Changed label to ‘Edge bent after meeting gravel’.

Comments and Suggestions No.71L775 Change to ‘the same sediment borehole guided by the lead ropes’

AnswerRevised

L775 Changed ’the same sediment borehole through the lead rope’ to ‘the same sediment borehole guided by the lead ropes’

Comments and Suggestions No.72Consider adding ‘, though hole refreezing and freezing in of the lead ropes, particularly in deep holes, remains a likely problem.’

AnswerRevised

Yes, the ‘hole refreezing and freezing in of the lead rope’ should be considered as the main technical challenge before such continuous subglacial coring system development. Sentence ‘Though hole refreezing and freezing in of the lead ropes, particularly in deep holes, remains a likely problem.’ has been added after ‘the same sediment borehole guided by the lead ropes.’

Comments and Suggestions No.73Additional BAS/UWITEC corer deployments and sediment core recoveries, Keith Makinson, BAS, UK.

12/13 Pine Island Glacier, three sites, ice thickness ~500-600 m, WCT ~200-400 m

Corer: BAS/UWITEC Main corer

Length of recovered cores: 30 cm, 92 cm, 115 cm

https://www.waisworkshop.org/sites/waisworkshop.org/files/files/agendas/2013/abstracts/Smith_J.pdf 

Smith, J. A., et al. (2016), Sub-ice-shelf sediments record history of twentieth-century retreat of Pine Island Glacier, Nature, advance online publication, doi:10.1038/nature20136.

2016/17 FNE1 Ice thickness 615 m, WCT 440 m

Latitude: 78.54451S, Longitude: 37.26143W

Corer: BAS/UWITEC Mini-corer

Length of recovered core: 64 cm

2016/17 FNE2 Ice thickness 597 m, WCT 588 m

Latitude: 78.56411S, Longitude: 38.08757W

Corer: BAS/UWITEC Mini-corer

Length of recovered core: 75.5 cm

Corer: BAS/UWITEC Main corer

Length of recovered core: 135.5 cm

2016/17 FNE3 Ice thickness 597 m, WCT 643 m

Latitude: 78.55305S, Longitude: 38.81890W

Corer: BAS/UWITEC Mini-corer

Length of recovered core: 75.5 cm

Corer: BAS/UWITEC Main corer

Length of recovered core: 152 cm

18/19 Rutford Ice Stream BEAMISH project.

Ice thickness 2152 m, the deepest ever hot water drilled subglacial access hole.

Latitude: 78.14S , Longitude: 83.92W

Corer: BAS/UWITEC Main corer

Length of recovered core: ~10 cm coarse gravel

Attempted coring at second site recovered no sediments.

Other sediments collected but with hot water drill nozzle sample cups. (Makinson and Anker, 2014)

https://www.bbc.co.uk/news/science-environment-46978496 

AnswerRevised

Thank you for providing all the above coring information.

All related parts were update based on ‘Additional BAS/UWITEC corer deployments and sediment core recoveries, Keith Makinson, BAS, UK.’, include:

(1) Text section description;

(2) Table 1;

(3) Figure 3;

(4) Figure 33.

References are added as following

62. Smith, J.; Shortt, M.; Bindschadler, B.; et al. A PIG unknown: the first sediment cores recovered from beneath. Available online: https://www.waisworkshop.org/sites/waisworkshop.org/files/files/agendas/2013/abstracts/Smith_J.pdf (accessed on 15 June 2019).

63. Smith, J.; Andersen, M.; Gaffney, A.; et al. Sub-ice-shelf sediments record history of twentieth-century retreat of Pine Island Glacier, Nature 2016, 541(7635), 77.

64. UK team drills record West Antarctic hole. Available online: https://www.bbc.com/news/science-environment-46978496 (accessed on 15 June 2019).

Thank you for your detailed comments and suggestions, I revised the paper based on all above comments and suggestions, please check.

Additional revisions:

(1) Added several references and updated reference number.

(2) In the author list, changed co-author name ‘Emma Smith’ to ‘Emma C. Smith’. This revision is require of co-author Emma Smith, please note that.

Reviewer 2 Report

 The authors should reduce the description of coring of antarctic subglacial  sediments in Abstract Section. 

 The authors should put maps and texts in the same page for Fig. 2 Structure and function schemes of the conventional corers.

The authors should more explain in coring methods for obtaining long cores in details.

The authors should add more citations into Journal of Marine Science and Engineering about coring of antarctic subglacial  sediments.  

Author Response

Detail response for Review report 2

Comments and Suggestions No.1             The authors should reduce the description of coring of Antarctic subglacial sediments in Abstract Section.

Answer Revised

                Reduced the description of coring of Antarctic subglacial sediments in Abstract Section by replaced sentence ‘because it can provide data from periods even earlier than the ice core, which are restricted to the age of the ice itself’ by ‘because it can provide data from periods even earlier than ice cores’.

Comments and Suggestions No.2             The authors should put maps and texts in the same page for Fig. 2 Structure and function schemes of the conventional corers.

Answer Revised

                We adjusted size of Fig 1 and Fig 2, the maps and texts are in the same page for Fig 2 now.

Comments and Suggestions

No.3      The authors should more explain in coring methods for obtaining long cores in details.

Answer Revised

                We added sentences to describe the coring methods for obtaining long cores more detailed. The sentences were added at the end of first paragraph of chapter ‘3.1.2. Coring methods for obtaining long cores’, as following:

‘... can help the core barrel go deeper into sediments. These power units are cylindrical shape designed to cope with radial size limitation caused by access borehole diameter. Corers for obtaining long sediment cores were equipped with increased potential energy (e.g. heavier gravity corer), continuous kinetic energy input (e.g. hammer/percussion corer) or penetration resistance reducing mechanism (e.g. vibrocroer).’

Comments and Suggestions No.4             The authors should add more citations into Journal of Marine Science and Engineering about coring of Antarctic subglacial sediments. 

Answer Revised

                We added citations from Journal of Marine Science and Engineering in to the Reference as following:

17. Matthew, S.; Oda, R.; Hilde, S.; et al. Projected 21st Century Sea-Level Changes, Observed Sea Level Extremes, and Sea Level Allowances for Norway. Journal of Marine Science and Engineering 2017, 5, 36. DOI: 10.3390/jmse5030036.

20. Thomas, G.; Paulus, P. Biorock Electric Reefs Grow Back Severely Eroded Beaches in Months. Journal of Marine Science and Engineering 2017, 5, 48. DOI: 10.3390/jmse5040048.

Thank you for your detailed comments and suggestions, I revised the paper based on all above comments and suggestions, please check.

Additional revisions:

(1) Added several references and updated reference number.

(2) In the author list, changed co-author name ‘Emma Smith’ to ‘Emma C. Smith’. This revision is require of co-author Emma Smith, please note that.

Reviewer 3 Report

The paper is an overview on the actual coring tools, focused on the working principles, operational methods, specific conditions and features, technical results, possible ways of improvements and future prospective in subglacial sediment coring approaches and corer design and development. It is a very extensive, complex and well documented study, with useful results for designers and practitioners in this field!

The following issues are recommended to improve the paper:

1.      The sub-ice streams principle in not clearly emphasizes in Fig. 1. Please clarify it!

2.      L. 88: “West Antarctica was ice-free was…” unclear statement. Idem line 469: “This corer is mounted on the on the front..”

3.      L. 409: It consists of five main parts: a control interface, hammer, core barrel and piston-clutch”. It seems there are only 4 parts!

Author Response

Detail response for Review report 3

The paper is an overview on the actual coring tools, focused on the working principles, operational methods, specific conditions and features, technical results, possible ways of improvements and future prospective in subglacial sediment coring approaches and corer design and development. It is a very extensive, complex and well documented study, with useful results for designers and practitioners in this field!

The following issues are recommended to improve the paper:

Comments and Suggestions No.1             The sub-ice streams principle in not clearly emphasizes in Fig. 1. Please clarify it!

Answer Revised

                Indeed, the sub-ice streams principle in not clearly emphasizes in Fig. 1. Therefore, we remade this the Fig.1 based on the Review report 1 with following corrections:

Changed ‘Basal ice flow’ to ‘Basal ice’.

Changed the central figure panel to ice stream, removed the lake.

Deleted other flow narrows, only left the flow narrows at the ice-sediment interface.

Comments and Suggestions No.2             L. 88: “West Antarctica was ice-free was…” unclear statement. Idem line 469: “This corer is mounted on the on the front..”

Answer Revised

                Deleted redundant ‘was’ in sentence ‘…, whereas the sedimentary records show that the last time central West Antarctica was ice-free was ~1 million years ago [17].’

Deleted redundant ‘on the’ in sentence ‘This corer is mounted on the on the front (lower end) of the Lake Ellsworth Probe which is sterilized before deployment and samples are capped upon retrieval (Fig. 21) [7].’

Comments and Suggestions No.3             L. 409: It consists of five main parts: a control interface, hammer, core barrel and piston-clutch”. It seems there are only 4 parts!

Answer Revised

                Added ‘piston’ in to this sentence and now it describes as following:

‘It consists of five main parts: a control interface, hammer, core barrel, piston, and piston-clutch [8].’

Thank you for your detailed comments and suggestions, I revised the paper based on all above comments and suggestions, please check.

Additional revisions:

(1) Added several references and updated reference number.

(2) In the author list, changed co-author name ‘Emma Smith’ to ‘Emma C. Smith’. This revision is require of co-author Emma Smith, please note that.

Round  2

Reviewer 2 Report

This paper has been revised by the authors according to the reviewer's comments. This paper should be accepted.